# HIERARCHICAL FUSION WITH DUAL CONTRAST FOR INCOMPLETE MULTI-VIEW REPRESENTATION LEARNING

## ABSTRACT

Deep incomplete multi-view clustering aims to learn consistent and discriminative representations from partially missing multi-view data, serving as an important research direction in multimodal learning. Existing approaches face two key limitations: (1) rigid fusion strategies that fail to dynamically select complementary views, and (2) contrastive learning methods that struggle to capture high-order dependencies and suppress intra-view redundancy. To address these issues, we propose DFHDC, a novel framework integrating dendritic fusion and hierarchical dual contrast mechanisms to dynamically select optimal view combinations and construct multi-level semantic fusion pathways. The dendritic fusion strategy progressively fuses views in a bottom-up manner to maximize inter-view complementarity, while the hierarchical dual contrast mechanism performs contrastive learning in both local and global semantic spaces, simultaneously maximizing cross-view mutual information and minimizing intra-view redundancy, thereby enhancing the consistency and discriminability of the learned representations. Additionally, the framework incorporates a view-specific fine-tuning strategy to implicitly recover missing views. Experiments show DFHDC outperforms state-of-the-art methods, especially under high missing rates, validating its effectiveness in incomplete multi-view learning.

## 1 INTRODUCTION

Deep learning techniques, with their capabilities in nonlinear feature extraction and representation learning, have become central to Deep Incomplete Multi-view Clustering (DIMC) (Wen et al., 2022). DIMC methods aim to extract informative representations, impute missing data, and achieve cross-view consistency alignment. Key technical paradigms include AutoEncoders (AEs) (Xu et al., 2023b) (reconstruction), Generative Adversarial Networks (GANs) (de Mello et al., 2022) (generation), Graph Convolutional Networks (GCNs) (Wang et al., 2023) (structural modeling), and contrastive learning (Lin et al., 2023) (consistency learning). Contrastive learning is particularly crucial for DIMC due to its inherent ability to enforce cross-view consistency by uncovering latent information and suppressing redundant features via negative pairs, thereby enhancing clustering.

Contrastive learning methods can be categorized into two types based on the semantic level of their contrastive objectives: (1) Local Semantic Space Contrastive Learning (Chen et al., 2020; Yuan et al., 2025; Cai et al., 2025). This class of methods constructs contrastive objectives at the view-specific feature representation level, aiming to pull together the local representations of the same instance across different views while pushing apart representations of different instances. However, these approaches tend to be sensitive to noise and may overlook higher-level semantic structures. (2) Common Semantic Space Contrastive Learning (Xu et al., 2023a; 2022). This type of method focuses on constructing contrastive objectives within a shared semantic space obtained through multi-view information fusion. While local semantic contrastive learning captures fine-grained view-specific features, it lacks a global perspective; in contrast, common semantic contrastive learning improves consistency of the fused representations but may overlook fine-grained distinctions and local structural information between samples. To address these limitations, recent studies (Zhu et al., 2025; Yang et al., 2023b) have proposed dual-level contrastive strategies that jointly leverage local

and global semantic consistency, aiming to capture both intra-sample consistency and inter-sample distributional alignment.

Despite these advances, critical challenges remain in practical applications: First, many existing methods, whether implicitly or explicitly, adopt a fixed fusion order or simply fuse all views simultaneously, lacking a mechanism to dynamically optimize the selection of complementary views. Such approaches may fail to identify the optimal view combinations that maximize information gain or minimize redundancy—particularly when dealing with numerous or highly heterogeneous views. Even sequential fusion strategies (Chao et al., 2024; Zhu et al., 2025) often rely on heuristic or predefined paths, which could overlook more effective combinations and lead to suboptimal representations. Second, current multi-view learning methods primarily align cross-view representations by constructing positive/negative sample pairs (Wang et al., 2025; Li et al., 2023). However, such approaches often capture only superficial similarities, struggle to model high-order semantic dependencies between views, and exhibit heightened susceptibility to redundant noise. Though some studies attempt to enhance cross-view semantic consistency using mutual information criteria (Lin et al., 2021; Xu et al., 2023a), they still lack explicit suppression of intra-view redundancy. This oversight leads to information duplication and noise propagation, significantly undermining model robustness and discriminative power—particularly pronounced in complex scenarios with high missing-data rates or substantial noise.

This paper introduces DFHDC, a novel framework for incomplete multi-view representation learning, built upon two key components: Dendritic Fusion and Hierarchical Dual Contrast. The dendritic fusion strategy constructs a robust common semantic representation by modeling each view's latent representation as the root node of a fusion tree. Through layer-wise pairwise fusion, it maximizes the complementarity among views, effectively capturing both shared and view-specific information. Building upon this, we introduce a Hierarchical Dual Contrast mechanism: For inter-level contrast, it aligns feature representations across different hierarchies (e.g., low-level local features with high-level shared representations) to enhance semantic consistency; for intra-level local contrast, it enforces a MaxMI-MinRed (Maximize Mutual Information-Minimize Redundancy) constrained contrastive learning at specific levels (particularly low-level features). The framework also integrates missing-view handling, dynamically completing absent views based on multi-view collaborative information, further refined by a view-specific fine-tuning mechanism. Finally, the completed view data and the learned hierarchical latent representations are jointly optimized end-to-end through reconstruction and clustering objectives. The main contributions of this paper are summarized as follows:

- We propose a dendritic fusion strategy that, through a bottom-up, multi-layered cascade tree structure, dynamically selects optimal view-pair combinations while preserving low-level details during the construction of multi-level semantic fusion pathway, thus achieving more comprehensive fusion of multi-view information.

- We propose a novel intra-level optimization mechanism that simultaneously maximizes cross-view mutual information and minimizes intra-view feature redundancy, thereby enhancing semantic discriminability.

- We introduce a fine-tuning strategy into our end-to-end framework for missing-view recovery and multi-view fusion, where view-specific networks perform refined optimization of the completed representations, achieving implicit completion of missing views. Experimental results on multiple benchmark datasets validate the superiority of the proposed method.

## 2 RELATED WORK

Contrastive learning methods can be categorized into two types based on the semantic level of their contrastive objectives: Local semantic space contrastive learning constructs contrastive objectives within each view or latent representation space, treating augmented representations of the same instance as positive pairs and representations of different instances as negative pairs, emphasizing instance-level consistency and discriminability. Representative methods include SimCLR (Chen et al., 2020), which uses large batch sizes to obtain more negative samples; PMIMC (Yuan et al., 2025), which performs prototype-level contrast in each view's latent space; and IMCFL (Cai et al.,

2025), which constructs positive and negative pairs using noise-augmented samples and incorporates similarity graph constraints. However, these methods struggle to capture global semantic consistency across views and are prone to inter-view discrepancies and local noise interference.

Common semantic space contrastive learning constructs contrastive objectives based on the fused representations of multiple views, aiming to enhance global consistency and discriminative power across views. Methods such as APADC (Xu et al., 2023a), which performs contrastive learning in the shared space through feature projection and distribution alignment, and DIMVC (Xu et al., 2022), which fuses multi-view features to create a common semantic space for cluster-level contrastive learning, are typical examples. However, these methods may overlook fine-grained structures within individual views, and the quality of fusion significantly impacts performance, potentially leading to insufficient semantic alignment. Local and global contrastive learning combines contrastive strategies across views and the common semantic space, capturing multi-level semantic consistency. It employs a dual contrastive mechanism to integrate local details (e.g., alignment between views) and global information (e.g., alignment between view representations and the common semantic space). Representative methods include DCL (Zhu et al., 2025) and Dealmvc (Yang et al., 2023b), which adopt dual-branch contrastive learning frameworks to coordinate local feature alignment and global semantic consistency.

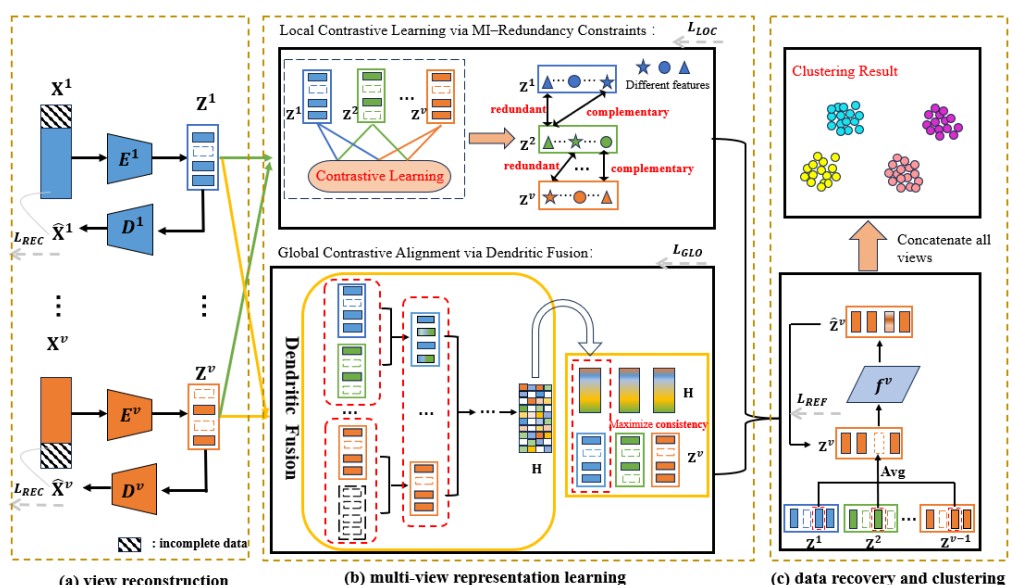

Figure 1: The DFHDC framework includes three modules: (a) view reconstruction, (b) multi-view representation learning, and (c) data completion and clustering. It uses autoencoders for feature extraction and reconstruction, contrastive learning for cross-view consistency, dendritic fusion for common representation, and fine-tuning for missing view completion and clustering.

## 3  METHOD

As shown in Figure 1, given an incomplete multi-view dataset $\mathbf{X}^1, \mathbf{X}^2, \ldots, \mathbf{X}^v$, each view is first mapped to a corresponding latent representation $\mathbf{Z}^1, \mathbf{Z}^2, \ldots, \mathbf{Z}^v$ via independent autoencoders. The model then performs two-stage contrastive learning: In the first stage, a dual-constraint strategy, which maximizes mutual information and minimizes redundancy, is employed to perform pairwise contrastive learning across different views. This approach aims to extract both consistent and complementary semantic information. In the second stage, a dendritic fusion mechanism is employed, where an adaptive complementarity optimization criterion is used to actively select optimal complementary view pairs for parallel fusion. The fusion results are recursively used to update the view set and iteratively aggregate multi-view semantics until a common representation $\mathbf{H}$ is obtained. Next, the view-specific representations $\mathbf{Z}^v$ are aligned with the common representation $\mathbf{H}$ in distribution. Furthermore, based on the learned latent features, the model uses the mean of the available view rep-

resentations as an initial imputation for the missing ones, which are then refined via view-specific fine-tuning networks to generate pseudo-reconstructed representations $\hat{\mathbf{Z}}^v$. In this process, representation learning and data completion mutually reinforce each other, ultimately enhancing clustering performance.

In this paper, $\left\{\mathbf{X}^v \in \mathbb{R}^{N \times d_v}\right\}_{v=1}^V$ represents a multi-view dataset with $V$ views, where $\mathbf{X}^v = \{\mathbf{x}_1^v, ..., \mathbf{x}_l^v, ..., \mathbf{x}_N^v\}$, $d_v$ represents the feature dimension of the $v$-th view, and $\mathbf{x}_l^v$ denotes the $l$-th sample in the $v$-th view. Additionally, we define an indicator matrix $\mathbf{A} \in \{0,1\}^{N \times V}$, whose elements are defined as follows:

$$\mathbf{A}_{iv} = \begin{cases} 1, & \text{if the } i\text{-th sample is available in the } v\text{-th view} \\ 0, & \text{otherwise.} \end{cases} \tag{1}$$

In this matrix, each column represents the availability status of all instances in a specific view.

To fully exploit the structural characteristics of each individual view, we construct a corresponding autoencoder for each view, where $E^v$ and $D^v$ denote the encoder and decoder of the $v$-th view, respectively. By introducing a reconstruction task, we guide the effective learning of the latent representation for each view. Specifically, given the input $\mathbf{X}^v$, the encoder extracts the latent representation $\mathbf{Z}^v = E^v(\mathbf{X}^v) \in \mathbb{R}^{N \times D}$, which is then reconstructed by the decoder as $\hat{\mathbf{X}}^v = D^v(\mathbf{Z}^v)$. The reconstruction loss is calculated using Mean Squared Error (MSE), which measures the discrepancy between the original and reconstructed data, thereby promoting the effective learning of the latent space. The overall reconstruction loss is defined as:

$$\mathcal{L}_{\text{REC}} = \sum_{v=1}^V \left\| \mathbf{X}^v - D^v\left(E^v\left(\mathbf{X}^v\right)\right) \right\|^2. \tag{2}$$

## 3.1 Local Contrastive Learning via MI–Redundancy Constraints

Let the latent representation of the $i$-th sample in the $v$-th view be denoted as $\mathbf{z}_i^v \in \mathbb{R}^D$, where $z_{i,m}^v$ represents the $m$-th dimensional feature value. Based on this, we define the cross-view feature-level similarity between the $m$-th feature in view $v_1$ and the $n$-th feature in view $v_2$ as $p_{m,n}^{(i,v_1,v_2)} = z_{i,m}^{v_1} \cdot z_{i,n}^{v_2}$. Furthermore, by aggregating the similarity of all samples, a cross-view joint feature similarity matrix $\mathbf{P}^{(v_1,v_2)} \in \mathbb{R}^{D \times D}$ between views $v_1$ and $v_2$ can be constructed, where the $(m,n)$-th entry is defined as:

$$P_{m,n}^{(v_1,v_2)} = \frac{1}{N} \sum_{i=1}^N p_{m,n}^{(i,v_1,v_2)} = \frac{1}{N} \sum_{i=1}^N z_{i,m}^{v_1} \cdot z_{i,n}^{v_2}. \tag{3}$$

The matrix $\mathbf{P}^{(v_1,v_2)}$ is normalized so that all its elements are non-negative and sum to 1, thus ensuring a stable and valid computation of mutual information. By summing the elements of the joint probability distribution matrix $\mathbf{P}^{(v_1,v_2)}$ along rows and columns, respectively, we obtain the marginal probability vectors for views $v_1$ and $v_2$, denoted as $\mathbf{P}^{(v_1)}$ and $\mathbf{P}^{(v_2)}$. Specifically, the marginal probability of the $m$-th dimension is given by $P_m^{(v_1)} = \sum_n P_{m,n}^{(v_1,v_2)}$, and the marginal probability of the $n$-th dimension is given by $P_n^{(v_2)} = \sum_m P_{m,n}^{(v_1,v_2)}$. Based on these, the mutual information loss between views $v_1$ and $v_2$ is defined as:

$$\mathcal{L}_{\text{MI}}^{(v_1,v_2)} = - \sum_{m=1}^D \sum_{n=1}^D P_{m,n}^{(v_1,v_2)} \log \left( \frac{P_{m,n}^{(v_1,v_2)}}{P_m^{(v_1)} \cdot P_n^{(v_2)}} \right). \tag{4}$$

To further enhance the discriminability and disentanglement of the latent representations, we introduce a feature redundancy suppression term $\mathcal{L}_{\text{red}}^{(v_1,v_2)}$ for each view pair $(v_1,v_2)$. Specifically, we concatenate and normalize the latent representations of views $v_1$ and $v_2$, and compute the feature correlation matrix $\mathbf{R}^{(v_1,v_2)}$ based on the concatenated feature dimensions. This matrix characterizes the statistical dependencies among feature dimensions in the joint cross-view feature space, reflecting the redundancy in the latent representations. Based on this correlation matrix, we define the

redundancy loss for the view pair $(v_1, v_2)$ as:

$$\mathcal{L}_{\text{red}}^{(v_1,v_2)} = \frac{1}{D(D-1)} \sum_{m \neq n} R_{m,n}^{(v_1,v_2)} \cdot \log \left( \frac{R_{m,n}^{(v_1,v_2)}}{R_m^{(v_1,v_2)} \cdot R_n^{(v_1,v_2)} + \epsilon} + \epsilon \right). \tag{5}$$

where $\epsilon$ is a small constant for numerical stability. Finally, we control the weight of the feature redundancy loss by the hyperparameter $\lambda_{\text{red}}$, and compute a weighted sum of the mutual information loss and feature redundancy loss over all view pairs, resulting in the overall cross-view contrastive loss:

$$\mathcal{L}_{\text{LOC}} = \frac{2}{V(V-1)} \sum_{1 \leq v_1 < v_2 \leq V} \left( \mathcal{L}_{\text{MI}}^{(v_1,v_2)} + \lambda_{\text{red}} \cdot \mathcal{L}_{\text{red}}^{(v_1,v_2)} \right). \tag{6}$$

### 3.2 GLOBAL CONTRASTIVE ALIGNMENT VIA DENDRITIC FUSION

Given the initial set of multi-view latent representations:

$$\mathcal{Z}^{(0)} = \{\mathbf{Z}^1, \mathbf{Z}^2, \dots, \mathbf{Z}^v\}, \tag{7}$$

as illustrated in Figure 1, we define the dendritic fusion process as an iterative grouping and pairing procedure. At each fusion layer, optimal complementary view pairs are actively selected and fused in parallel according to a dynamic complementarity optimization criterion. This "selective pairing and parallel fusion" strategy ensures that each step in constructing the fusion tree progresses toward maximizing the overall complementary information. To effectively capture the complementary information between views, we define a fusion score function between any two representations $\mathbf{Z}_{(t)}^{v_1}$ and $\mathbf{Z}_{(t)}^{v_2}$ within the set $\mathcal{Z}^{(t)}$ as:

$$S\left(\mathbf{Z}_{(t)}^{v_1}, \mathbf{Z}_{(t)}^{v_2}\right) = 1 - \text{Sim}\left(\mathbf{Z}_{(t)}^{v_1}, \mathbf{Z}_{(t)}^{v_2}\right), \tag{8}$$

where $\text{Sim}(\cdot, \cdot)$ denotes the cosine similarity, which measures the proximity between two representations in the semantic space. This score function evaluates the complementarity between view representations by computing their dissimilarity, i.e., $1 - \text{Sim}(\cdot, \cdot)$, thereby promoting effective complementary interaction across views.

We introduce a binary variable $x_{v_1 v_2}^{(t)}$ to indicate whether the $v_1$-th and $v_2$-th representations are selected for fusion in iteration $t$, defined as:

$$x_{v_1 v_2}^{(t)} = \begin{cases} 1, & \text{if } \mathbf{Z}_{(t)}^{v_1} \text{ and } \mathbf{Z}_{(t)}^{v_2} \text{ are fused at iteration } t, \\ 0, & \text{otherwise.} \end{cases} \tag{9}$$

Based on this, the fusion decision process in the $t$-th iteration can be formulated as a combinatorial optimization problem, with the detailed solution provided in Appendix A.1.

$$\max_{\mathbf{X}^{(t)} \in \{0,1\}^{N_t \times N_t}} \sum_{v_1=1}^{N_t} \sum_{v_2=1}^{N_t} S\left(\mathbf{Z}_{(t)}^{(v_1)}, \mathbf{Z}_{(t)}^{(v_2)}\right) \cdot x_{v_1 v_2}^{(t)},$$

$$\text{s.t. } x_{v_1 v_1}^{(t)} = 0, \sum_{v_2=1}^{N_t} x_{v_1 v_2}^{(t)} \leq 1, \ x_{v_1 v_2}^{(t)} = x_{v_2 v_1}^{(t)}, \ x_{v_1 v_2}^{(t)} \in \{0,1\}, \forall v_1, v_2 \tag{10}$$

Here, $N_t$ denotes the size of the current set of latent representations at iteration $t$, and $\mathbf{X}^{(t)} \in \{0,1\}^{N_t \times N_t}$ is the pairing matrix indicating which view pairs $(v_1, v_2)$ are fused. To ensure the rationality of the fusion process, we impose the following constraints: each representation participates in at most one fusion operation per iteration to avoid reuse; additionally, to guarantee the validity of the pairing, self-fusion is prohibited and the fusion relation is required to be symmetric.

The objective of this optimization problem is to maximize the total fusion score of the selected view pairs in the current iteration by solving for the pairing matrix $\mathbf{X}^{(t)}$. For each selected pair $(v_1^*, v_2^*)$, a new fused representation is constructed by averaging the two, i.e.,

$$\mathbf{Z}_{(t)}^{v_1^*, v_2^*} = \frac{1}{2}\left(\mathbf{Z}_{(t)}^{v_1^*} + \mathbf{Z}_{(t)}^{v_2^*}\right). \tag{11}$$

After completing one iteration of fusion, the newly generated representations form the set of latent representations for the next iteration, i.e.,

$$\mathcal{Z}^{(t+1)} = \left\{ \mathbf{Z}_{(t)}^{v_1^*, v_2^*} \mid (v_1^*, v_2^*) \in \mathcal{P}^{(t)} \right\} \cup \left\{ \mathbf{Z}_{(t)}^{v} \mid v \notin \bigcup_{(v_1^*, v_2^*) \in \mathcal{P}^{(t)}} \{v_1^*, v_2^*\} \right\}, \quad (12)$$

where $\mathcal{P}^{(t)} = \{(v_1^*, v_2^*) \mid x_{v_1^* v_2^*}^{(t)} = 1\}$ denotes the set of all selected fusion pairs at iteration $t$. This fusion process is iteratively performed and eventually yields a unified common representation $\mathbf{H}$, i.e.,

$$\mathbf{H} = \mathcal{Z}^{(T)}. \quad (13)$$

The resulting representation integrates complementary information across different views, achieving both local specificity and global consistency. To further enhance the semantic alignment between the common representation and each individual view, we introduce a global contrastive loss to guide the learning of latent representations. Specifically, given two sets of representations, $\mathbf{Z}^v = \{\mathbf{z}_1^v, \mathbf{z}_2^v, \ldots, \mathbf{z}_N^v\}$ and $\mathbf{H} = \{\mathbf{h}_1, \mathbf{h}_2, \ldots, \mathbf{h}_N\}$, we define the contrastive loss between the $v$-th view and the common representation as:

$$\mathcal{L}_{\text{contrast}}^v (\mathbf{Z}^v, \mathbf{H}) = -\frac{1}{N} \sum_{i=1}^{N} \log \frac{\exp\left(\text{sim}\left(\mathbf{z}_i^v, \mathbf{h}_i\right)\right)}{\sum_{j=1, j\neq i}^{N} \exp\left(\text{sim}\left(\mathbf{z}_i^v, \mathbf{h}_j\right)\right)}, \quad (14)$$

where $\text{sim}(\mathbf{z}_i^v, \mathbf{h}_i)$ denotes the similarity between the representation vector $\mathbf{z}_i^v$ and the corresponding common representation $\mathbf{h}_i$. Finally, the global contrastive loss is obtained by aggregating the contrastive losses across all views, defined as:

$$\mathcal{L}_{\text{GLO}} = \frac{1}{V} \sum_{v=1}^{V} \mathcal{L}_{\text{contrast}}^v (\mathbf{Z}^v, \mathbf{H}). \quad (15)$$

### 3.3 FINE-TUNING FOR VIEW COMPLETION

In incomplete multi-view clustering, to address the issue of missing representations in certain views for some samples, we design a view-specific refinement module. Specifically, assuming that the $v_1$-th view is missing, its latent representation is first approximated by weighted aggregation of the representations from other available views, and then reconstructed through a view-specific fine-tuning network, defined as follows:

$$\hat{\mathbf{Z}}^{v_1} = f_{\text{refine}}^{v_1} \left( \frac{1}{V-1} \sum_{\substack{v_2=1 \\ v_2 \neq v_1}}^{V} \mathbf{Z}^{v_2} \right), \quad (16)$$

where $f_{\text{refine}}^v(\cdot)$ denotes the view-specific refinement network, which is designed to enhance the representation quality and inter-view consistency. To this end, we introduce a refinement loss based on the MSE, which supervises the discrepancy between the reconstructed and original representations for each view, defined as:

$$\mathcal{L}_{\text{REF}} = \frac{1}{V} \sum_{v_1=1}^{V} \left\| \hat{\mathbf{Z}}^{v_1} - \mathbf{Z}^{v_1} \right\|_F^2. \quad (17)$$

### 3.4 OBJECTIVE FUNCTION AND OPTIMIZATION ALGORITHM

Thus, our overall objective function can be expressed as:

$$\mathcal{L} = \lambda_1 \mathcal{L}_{\text{REC}} + \lambda_2 \mathcal{L}_{\text{LOC}} + \lambda_3 \mathcal{L}_{\text{GLO}} + \lambda_4 \mathcal{L}_{\text{REF}}. \quad (18)$$

The overall training procedure of DFHDC is summarized in Algorithm 1.

---

**Algorithm 1** Incomplete Multi-View Representation Learning via Dendritic Fusion and Hierarchical Dual Contrast

---

1: **Input:** Incomplete multi-view dataset $\{\mathbf{X}^v\}_{v=1}^V$ with indicator matrix $\mathbf{A}$; epochs $E$
2: **Parameters:** Trade-off coefficients $\lambda_1, \lambda_2, \lambda_3, \lambda_4$
3: **Output:** Clustering results
4: **while** epoch $\leq E$ **do**
5:     Learn view-specific representations via Eqs. (2) and (6)
6:     Compute fusion score $S$ via Eq. (8)
7:     Determine the pairing scheme with the highest total score via Eq. (10)
8:     Update current representation set $\mathcal{Z}^{(t)}$ via Eq. (12)
9:     Repeat until one common representation $\mathbf{H}$ remains
10:     Align $\mathbf{Z}^v$ and $\mathbf{H}$ via Eq. (15)
11:     Obtain $\hat{\mathbf{Z}}^{v_1}$ via Eq. (16) to complete the views
12:     Jointly optimize representations via Eq. (18)
13: **end while**
14: Perform $k$-means on concatenated views

---

## 4 EXPERIMENTS

### 4.1 DATASETS AND COMPARED METHODS

To evaluate the effectiveness of the proposed method, we selected five widely used and representative datasets in our experiments: Hdigit (Yang et al., 2021), which contains 10,000 samples, 2 views, with dimensionalities of 784 and 256; 100leaves (Zheng et al., 2022), which contains 1,600 samples, 3 views, each with a dimensionality of 64; Fashion (Zheng et al., 2018), which contains 10,000 samples, 3 views, each with a dimensionality of 784; MNIST_USPS (Peng et al., 2019), which contains 5,000 samples, 2 views, each with a dimensionality of 784; and HandWritten (LeCun et al., 1989), which contains 2,000 samples, 6 views, with dimensionalities of 240, 76, 216, 47, 64, and 6, respectively.

The comparison includes 10 state-of-the-art incomplete multi-view clustering methods: **IMC-MCL** (Yin et al., 2025) simultaneously optimizes missing data recovery and multi-view consistency. **MICA** (Wang et al., 2025) enhances the recovery quality of missing views and the consistency of cross-view clustering through multi-level imputation and contrastive alignment. **MRL_CAL** (Wang et al., 2024) combines contrastive learning and adversarial learning to extract multi-level features. **ICMVC** (Chao et al., 2024) introduces a confidence-guided contrastive learning strategy with instance-level attention to exploit multi-view consistency and complementarity. **APADC** (Xu et al., 2023a) employs autoencoders for view-specific feature learning and jointly optimizes mutual information and distribution alignment in a shared space. **MCAC** (Zhang & Zhu, 2023) leverages attention-based contrastive learning to enhance cross-view consistency. **DCP** (Lin et al., 2022) proposes a dual contrastive prediction framework with information-theoretic objectives for view recovery and consistency. **DSIMVC** (Tang & Liu, 2022) completes missing views based on semantic neighbors and selects imputed samples dynamically to mitigate semantic bias. **SURE** (Yang et al., 2023a) leverages cross-view positive pairs and introduces a noise-robust contrastive loss to mitigate the effect of negative samples. **COMPLETER** (Lin et al., 2021) integrates contrastive learning with bidirectional prediction to jointly learn consistent representations and recover missing views.

### 4.2 EXPERIMENTAL SETTING

All methods are evaluated using PyTorch 2.3.1 on a platform equipped with an NVIDIA RTX 4070 GPU. To ensure fairness, our method employs a unified fully connected autoencoder across all datasets (encoder: *Input*–1024–1024–1024–*Output*, symmetric decoder: *Output*–1024–1024–1024–*Input*, with ReLU activations). During fine-tuning, each view uses a two-layer fully connected network with a ReLU activation in between to ensure non-linear representation.

Each dataset is trained for $E$ epochs (e.g., $E = 500$) with a learning rate of 0.0001 and a batch size of 256. Comparative experiments are conducted under missing rates of 0.1, 0.3, and 0.5, as well as

Table 1: Performance comparison of different methods on various datasets

| Setting | Method | Hdigit | | | 100leaves | | | Fashion | | | MNIST_USPS | | | HandWritten | | | YouTubeFace | | |
|---|---|---|---|---|---|---|---|---|---|---|---|---|---|---|---|---|---|---|---|
| | | ACC | NMI | ARI | ACC | NMI | ARI | ACC | NMI | ARI | ACC | NMI | ARI | ACC | NMI | ARI | ACC | NMI | ARI |
| Partially | IMC-MCL | 97.35 | 92.84 | 94.21 | 24.16 | 60.40 | 15.92 | 66.11 | 77.00 | 59.98 | 80.82 | 85.36 | 77.32 | 69.95 | 62.61 | 52.38 | 12.62 | 7.91 | 1.31 |
| | MICA | 58.86 | 72.52 | 61.93 | 9.25 | 29.25 | 1.55 | 88.16 | 85.11 | 82.73 | 58.68 | 73.73 | 59.71 | 29.25 | 38.12 | 16.99 | 21.39 | 4.19 | 0.33 |
| | MRL_CAL | 12.62 | 0.82 | 0.28 | 12.12 | 49.35 | 7.03 | 82.29 | 81.39 | 75.62 | 91.58 | 81.52 | 82.02 | 48.00 | 54.52 | 40.87 | 12.14 | 9.06 | 1.55 |
| | ICMVC | 16.44 | 8.67 | 3.64 | 50.10 | 74.06 | 30.89 | 77.89 | 73.64 | 66.76 | 96.07 | 90.53 | 91.47 | 52.57 | 50.77 | 34.78 | 16.81 | 7.23 | 3.99 |
| | APADC | 91.50 | 81.70 | 82.00 | 30.50 | 65.00 | 18.80 | 68.70 | 76.90 | 61.50 | 95.30 | 89.30 | 89.80 | 73.50 | 70.30 | 58.10 | 23.14 | 4.17 | 1.04 |
| | MCAC | 24.08 | 14.75 | 6.93 | 27.59 | 58.43 | 12.14 | 53.59 | 57.51 | 39.01 | 71.06 | 64.03 | 58.33 | 40.92 | 37.34 | 22.04 | 13.36 | 11.92 | 3.92 |
| | DCP | 92.15 | 88.84 | 85.03 | 42.75 | 70.63 | 28.22 | 86.29 | 81.72 | 76.11 | 86.61 | 85.56 | 78.34 | 65.43 | 70.29 | 44.32 | 23.07 | 19.58 | 3.76 |
| | DSIMVC | 94.40 | 89.37 | 90.40 | 27.29 | 59.14 | 15.10 | 87.03 | 81.67 | 77.16 | 96.73 | 91.81 | 92.89 | 70.14 | 68.36 | 57.46 | 15.06 | 5.24 | 0.50 |
| | SURE | 66.59 | 17.66 | 15.47 | 30.52 | 58.81 | 10.46 | 83.80 | 77.85 | 71.92 | 78.15 | 74.46 | 67.09 | 71.63 | 60.19 | 53.86 | 11.96 | 9.14 | 2.81 |
| | COMP | 91.36 | 80.73 | 81.85 | 28.40 | 62.32 | 15.68 | 79.89 | 78.48 | 69.86 | 95.61 | 90.44 | 90.72 | 65.65 | 68.38 | 43.87 | 18.25 | 12.28 | 1.69 |
| | **DFHDC** | **97.74** | **93.82** | **95.05** | **53.21** | **74.72** | **33.77** | **93.21** | **87.35** | **86.28** | **97.06** | **92.49** | **93.59** | **75.60** | **72.92** | **60.18** | **35.26** | **32.66** | **7.95** |
| Fully | IMC-MCL | 99.05 | 97.18 | 97.05 | 24.24 | 65.59 | 18.63 | 75.89 | 85.82 | 71.69 | 76.06 | 87.54 | 73.27 | 73.89 | 67.55 | 58.27 | 15.55 | 10.15 | 2.07 |
| | MICA | 99.03 | 96.97 | 97.23 | 11.13 | 34.76 | 1.71 | 97.25 | 94.59 | 94.24 | 98.56 | 97.18 | 97.03 | 20.90 | 34.94 | 9.87 | 26.01 | 0.67 | 0.17 |
| | MRL_CAL | 12.97 | 0.74 | 0.27 | 14.31 | 52.77 | 8.43 | 94.21 | 90.59 | 88.44 | 98.66 | 96.33 | 97.05 | 56.45 | 60.80 | 44.03 | 14.17 | 12.91 | 2.34 |
| | ICMVC | 16.40 | 8.59 | 3.48 | 16.80 | 78.38 | 25.52 | 92.43 | 87.10 | 85.12 | 99.01 | 97.28 | 97.80 | 20.76 | 12.54 | 4.96 | 17.28 | 8.01 | 4.12 |
| | APADC | 85.10 | 73.90 | 70.40 | 67.40 | 85.70 | 59.00 | 76.10 | 81.90 | 64.60 | 98.24 | 95.70 | 96.10 | 72.00 | 67.40 | 55.50 | 22.19 | 2.68 | 1.03 |
| | MCAC | 29.52 | 27.08 | 13.01 | 36.70 | 68.75 | 22.09 | 39.50 | 54.20 | 19.04 | 98.43 | 97.13 | 97.40 | 73.87 | 70.02 | 59.73 | 13.81 | 14.48 | 5.08 |
| | DCP | 96.82 | 96.59 | 95.00 | 55.90 | 84.96 | 48.95 | 92.92 | 90.22 | 86.59 | 90.32 | 91.40 | 84.68 | 66.65 | 74.15 | 49.33 | 25.75 | 27.67 | 6.68 |
| | DSIMVC | 91.66 | 90.34 | 88.46 | 32.74 | 66.31 | 21.00 | 69.86 | 69.73 | 58.12 | 98.79 | 96.69 | 97.34 | 70.73 | 70.11 | 59.66 | 15.92 | 6.40 | 1.25 |
| | SURE | 78.72 | 69.81 | 65.23 | 51.78 | 79.50 | 38.26 | 90.80 | 87.84 | 83.79 | 98.89 | 97.12 | 97.88 | 57.66 | 51.29 | 38.89 | 13.15 | 10.42 | 3.31 |
| | COMP | 95.43 | 89.76 | 90.25 | 42.54 | 77.95 | 34.39 | 84.00 | 87.07 | 79.73 | 97.84 | 95.74 | 95.31 | 67.45 | 70.95 | 46.67 | 18.87 | 17.59 | 3.57 |
| | **DFHDC** | **99.13** | **97.22** | **97.27** | **74.13** | **90.08** | **67.40** | **98.01** | **96.07** | **95.77** | **99.06** | **97.38** | **97.93** | **93.06** | **88.74** | **84.78** | **39.75** | **40.69** | **9.25** |

with complete views. Results for missing rate 0.5 and complete views are reported in the main text, while other results are provided in Appendix B.1.

## 4.3 EXPERIMENTAL RESULTS AND ANALYSIS

We compared our method with state-of-the-art approaches on five benchmark datasets under complete views, 50% missing views, and Gaussian noise conditions. The results in Tables 1 and 2 show that our method consistently achieves strong performance across different datasets and experimental settings. The best and second-best results are highlighted in **bold** and underlined, respectively.

**(Complete Incomplete Views)** The experimental results in Table 1 show that our method outperforms the second-best IMC-MCL on the Hdigit dataset in all three metrics. On the Fashion dataset, it significantly outperforms MICA with an ACC of 93.21%. For the 100leaves dataset, which has a large number of classes and complex data structures, our method still demonstrates strong discriminative power, surpassing the second-best ICMVC in all three metrics. On the HandWritten dataset, our method achieves an ACC of 93.06% under the complete data setting, significantly outperforming the second-best IMC-MCL, and also demonstrates clear superiority over other comparison methods under the incomplete view setting. Finally, on the MNIST_USPS dataset, our method attains excellent performance across all three clustering metrics, outperforming all comparison algorithms. Moreover, we further conducted experiments on the large-scale YouTubeFace dataset, where the proposed method still achieves the best performance among all the compared methods.

**(Gaussian Noise)** The experimental results are shown in Table 2. Under Gaussian noise conditions, our method still demonstrates strong robustness. On the Hdigit dataset, DFHDC maintains stable performance across all three clustering metrics, achieving an accuracy of 97.87%. On the 100leaves and MNIST_USPS datasets, DFHDC outperforms other methods in ACC, NMI, and ARI. Finally, on the HandWritten dataset, even with added noise, DFHDC achieves an ACC of 76.38%, and its ARI improves by 5.5% compared to the second-best method DSIMVC (53.30%).

## 4.4 SCALABILITY ANALYSIS

To evaluate the scalability of the proposed method, we conducted experiments under two settings: varying the number of views with a fixed number of samples, and varying the number of samples with a fixed number of views. In both cases, we recorded clustering performance metrics (ACC, NMI, and ARI), running time, and memory consumption.

Table 2: Performance comparison of different methods under Gaussian noise

| Setting | Method | Hdigit | | | 100leaves | | | Fashion | | | MNIST_USPS | | | HandWritten | | |
|---------|--------|------|------|------|------|------|------|------|------|------|------|------|------|------|------|------|
| | | ACC | NMI | ARI | ACC | NMI | ARI | ACC | NMI | ARI | ACC | NMI | ARI | ACC | NMI | ARI |
| Noise | IMC-MCL | 97.04 | 92.18 | 93.54 | 24.31 | 59.63 | 15.51 | 67.69 | 78.00 | 61.63 | 79.12 | 83.59 | 76.21 | 68.70 | 61.40 | 50.65 |
| | MICA | 49.59 | 66.12 | 50.28 | 8.38 | 29.02 | 0.75 | 87.73 | 85.32 | 81.81 | 48.76 | 68.29 | 50.62 | 19.05 | 17.26 | 8.48 |
| | MRL_CAL | 15.96 | 4.26 | 1.92 | 10.37 | 49.07 | 7.05 | 82.29 | 81.39 | 75.62 | 91.30 | 80.77 | 81.65 | 49.90 | 53.32 | 37.79 |
| | ICMVC | 16.69 | 10.02 | 3.93 | 47.68 | 72.79 | 28.55 | 75.79 | 72.88 | 65.26 | 93.94 | 88.43 | 88.22 | 49.26 | 46.91 | 32.70 |
| | APADC | 90.19 | 79.77 | 79.16 | 31.50 | 63.30 | 13.20 | 65.34 | 70.09 | 55.08 | 93.50 | 86.20 | 86.10 | 71.10 | 70.90 | 53.00 |
| | MCAC | 27.83 | 20.49 | 9.30 | 27.48 | 58.68 | 12.23 | 85.55 | 78.54 | 74.70 | 93.14 | 85.51 | 85.51 | 43.18 | 41.09 | 23.44 |
| | DCP | 97.68 | 93.62 | 94.92 | 43.36 | 70.20 | 28.21 | 90.08 | 83.25 | 80.65 | 82.38 | 82.75 | 73.64 | 67.31 | 71.68 | 46.40 |
| | DSIMVC | 93.61 | 87.12 | 88.05 | 25.87 | 58.39 | 14.29 | 83.48 | 77.37 | 71.40 | 92.06 | 88.52 | 85.33 | 74.40 | 71.27 | 53.30 |
| | SURE | 48.46 | 34.25 | 25.16 | 31.36 | 59.37 | 11.55 | 85.08 | 77.91 | 73.10 | 70.08 | 65.42 | 56.39 | 67.16 | 59.13 | 50.24 |
| | COMP | 49.39 | 47.76 | 31.49 | 31.64 | 63.21 | 18.11 | 78.68 | 77.72 | 67.94 | 87.93 | 83.63 | 78.18 | 66.44 | 68.80 | 42.74 |
| | **DFHDC** | **97.87** | **94.06** | **95.33** | **50.48** | **73.86** | **31.71** | **92.11** | **86.39** | **84.36** | **94.58** | **88.88** | **89.00** | **76.38** | **72.19** | **58.80** |

Table 3: Loss ablation experiments on three datasets with the missing rate of 0.5

| Model | Loss Components | | | | Hdigit | | | 100leaves | | | HandWritten | | |
|-------|-------|-------|-------|-------|------|------|------|------|------|------|------|------|------|
| | $\mathcal{L}_{REC}$ | $\mathcal{L}_{LOC}$ | $\mathcal{L}_{GLO}$ | $\mathcal{L}_{REF}$ | ACC | NMI | ARI | ACC | NMI | ARI | ACC | NMI | ARI |
| M-1 | ✓ | | | | 27.65 | 27.43 | 12.23 | 26.87 | 57.96 | 13.11 | 19.14 | 12.99 | 7.03 |
| M-2 | | ✓ | | | 75.27 | 73.80 | 54.10 | 41.64 | 66.14 | 15.91 | 51.80 | 51.20 | 23.70 |
| M-3 | | | ✓ | | 46.57 | 62.55 | 44.77 | 22.29 | 54.79 | 8.78 | 43.28 | 46.35 | 25.21 |
| M-4 | | | | ✓ | 12.06 | 1.73 | 1.17 | 23.01 | 54.59 | 9.59 | 55.98 | 50.54 | 32.34 |
| M-5 | | ✓ | | ✓ | 49.46 | 69.50 | 49.13 | 21.85 | 54.42 | 8.32 | 53.07 | 58.16 | 40.83 |
| M-6 | ✓ | | ✓ | | 49.28 | 69.95 | 49.06 | 22.99 | 54.65 | 10.90 | 28.80 | 29.82 | 16.48 |
| M-7 | ✓ | | | ✓ | 58.01 | 57.42 | 44.55 | 35.93 | 64.35 | 19.05 | 45.66 | 46.29 | 24.89 |
| M-8 | | ✓ | ✓ | | 72.35 | 69.41 | 47.12 | 47.49 | 67.62 | 17.45 | 53.34 | 53.49 | 26.01 |
| M-9 | ✓ | ✓ | | | 75.58 | 70.38 | 50.73 | 48.04 | 68.13 | 18.85 | 41.44 | 39.60 | 22.83 |
| M-10 | | ✓ | | ✓ | 96.93 | 92.30 | 94.47 | 52.27 | 72.90 | 31.94 | 62.73 | 66.03 | 35.34 |
| M-11 | ✓ | | ✓ | ✓ | 68.57 | 73.81 | 62.76 | 22.04 | 54.66 | 9.45 | 44.06 | 43.93 | 26.31 |
| M-12 | ✓ | ✓ | ✓ | | 73.35 | 70.42 | 48.64 | 46.94 | 67.41 | 17.42 | 39.17 | 37.53 | 22.46 |
| M-13 | | ✓ | ✓ | ✓ | 97.43 | 92.90 | 94.79 | 53.11 | 73.65 | 33.24 | 67.16 | 69.08 | 41.92 |
| M-14 | ✓ | ✓ | | ✓ | 97.60 | 93.76 | 94.78 | 53.18 | 73.59 | 30.96 | 72.01 | 70.89 | 54.24 |
| DFHDC | ✓ | ✓ | ✓ | ✓ | **97.74** | **93.82** | **95.05** | **53.21** | **74.22** | **33.77** | **75.60** | **72.92** | **60.18** |

As shown in Table 4, on the HandWritten dataset, when the number of views increases from 2 to 6, the training time rises from 67.19 s to 384.39 s, while the clustering performance continues to improve, indicating that the proposed method exhibits good scalability in multi-view scenarios.

As shown in Table 5, on the Fashion dataset, as the number of samples increases from 2,000 to 10,000, the clustering performance steadily improves. Although the running time and memory consumption increase accordingly, the overall growth remains moderate, which further demonstrates the scalability and stability of the proposed method on large-scale data.

Table 4: Performance of DFHDC on HandWritten dataset with different numbers of views.

| Views Number | ACC | NMI | ARI | Time (s) | Memory (MB) |
|--------------|-----|-----|-----|----------|-------------|
| 2 | 53.57 | 53.84 | 30.35 | 67.19 | 198.51 |
| 3 | 57.52 | 55.56 | 35.25 | 165.45 | 293.48 |
| 4 | 64.34 | 60.78 | 49.34 | 247.23 | 378.31 |
| 5 | 70.89 | 65.89 | 55.65 | 273.12 | 464.14 |
| 6 | 75.60 | 72.92 | 60.18 | 384.39 | 546.81 |

## 4.5 ABLATION STUDY

To investigate the effectiveness of each loss component, we conducted ablation studies on three datasets: Hdigit, 100leaves, and HandWritten, evaluating four components: $\mathcal{L}_{REC}$, $\mathcal{L}_{LOC}$, $\mathcal{L}_{GLO}$, and $\mathcal{L}_{REF}$. As shown in Table 3, the $\mathcal{L}_{LOC}$ component alone achieves an accuracy of 75.27% on the Hdigit dataset, while models lacking this component exhibit significantly degraded performance, demonstrating its critical role in representation learning. On the HandWritten dataset, which contains more views, the M-13 variant incorporating our proposed dendritic fusion mechanism clearly

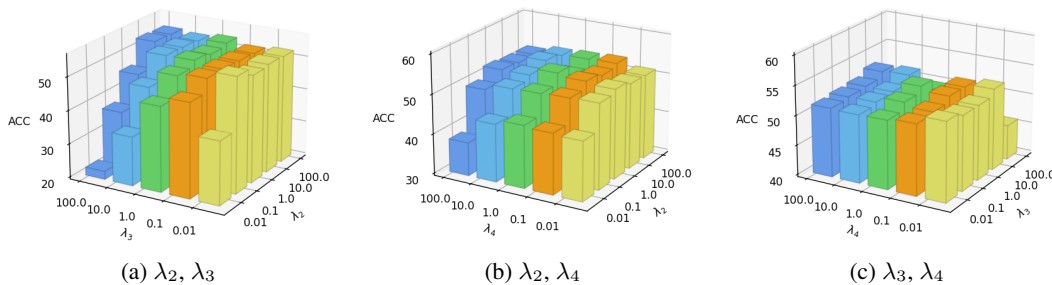

(a) $\lambda_2, \lambda_3$         (b) $\lambda_2, \lambda_4$         (c) $\lambda_3, \lambda_4$

Figure 2: Parameter sensitivity analysis on the 100leaves dataset with a missing rate of 0.5 and $\lambda_1 = 0.1$.

Table 5: Performance of DFHDC on the Fashion dataset with different sample sizes.

| Sample Ratio | Number of Samples | ACC | NMI | ARI | Time (s) | Memory (MB) |
|---|---|---|---|---|---|---|
| 0.2 | 2000 | 79.07 | 78.42 | 66.70 | 167.96 | 398.20 |
| 0.4 | 4000 | 83.02 | 80.05 | 71.21 | 330.38 | 437.23 |
| 0.6 | 6000 | 86.67 | 83.36 | 77.09 | 495.80 | 476.76 |
| 0.8 | 8000 | 89.66 | 85.07 | 81.66 | 607.37 | 514.32 |
| 1.0 | 10000 | 93.21 | 87.35 | 86.28 | 888.92 | 568.05 |

outperforms M-10, which does not use this strategy. Additionally, on the 100leaves dataset, M-8 (which integrates both $\mathcal{L}_{\text{LOC}}$ and $\mathcal{L}_{\text{GLO}}$) significantly outperforms models using a single loss term (M-2 or M-3), verifying the necessity of the hierarchical dual contrastive learning design. Finally, comprehensive analysis of the ablation results shows that the full-optimization model DFHDC achieves the best performance across all datasets, fully validating the effectiveness of our dendritic fusion, hierarchical dual contrast, and fine-tuned view completion strategies in complex multi-view clustering tasks. Additional ablation results on other datasets are provided in Appendix B.2.

### 4.6 PARAMETER ANALYSIS

In this section, we conduct a sensitivity analysis of the four hyperparameters $\lambda_1$, $\lambda_2$, $\lambda_3$, and $\lambda_4$ in the total loss function of our method. On the 100leaves dataset, we fix the weight of the reconstruction loss $\mathcal{L}_{\text{REC}}$ to 0.1 and set the remaining hyperparameters within the range $\{0.01, 0.1, 1, 10, 100\}$ for combination experiments. The experimental results are shown in Figure 2. From the figure, it can be observed that $\lambda_2$ plays a crucial bridging role in the model. A moderate local contrastive loss can significantly improve the model's performance in handling missing views. Moreover, the sensitivity analysis of $\lambda_3$ and $\lambda_4$ indicates that a balanced setting between them is essential for achieving optimal performance. A significant discrepancy between their values leads to a notable performance drop. The results on the other datasets are provided in Appendix B.3. Convergence analysis and visualizations are provided in Appendix B.4 and Appendix B.5

## 5 CONCLUSION

This paper proposes DFHDC, a novel framework for incomplete multi-view representation learning. The framework employs a dendritic fusion strategy to dynamically select optimal view pairings and construct a multi-level semantic fusion pathway. It integrates a dual contrast mechanism to enhance the consistency and discriminability of semantic representations. A view-specific fine-tuning strategy is also incorporated to refine missing views. Experimental results demonstrate the superior performance of the proposed method in handling missing-view tasks. It is worth noting that the dendritic fusion strategy has a time complexity of $O(V^3)$, which may pose computational challenges when dealing with large-scale data. Future work will focus on improving computational efficiency.

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

## A  Optimization Problem Analysis

### A.1  Solving the Optimization Problem

To solve the view pairing optimization problem, which aims to select the optimal pairing configuration from the representation set $\mathcal{Z}^{(t)}$ at the $t$-th iteration to maximize the overall fusion complementarity score, we reformulate it as a standard minimum-cost bipartite matching problem and apply the Hungarian algorithm for efficient solution. First, we define the pairing score function as:

$$S\left(\mathbf{Z}_{(t)}^{v_1}, \mathbf{Z}_{(t)}^{v_2}\right) = 1 - \text{Sim}\left(\mathbf{Z}_{(t)}^{v_1}, \mathbf{Z}_{(t)}^{v_2}\right), \tag{19}$$

we construct the corresponding cost matrix $\mathbf{A}^{(t)} \in \mathbb{R}^{N_t \times N_t}$, with its elements defined as:

$$A_{v_1 v_2}^{(t)} = -S\left(\mathbf{Z}_{(t)}^{v_1}, \mathbf{Z}_{(t)}^{v_2}\right) = \text{Sim}\left(\mathbf{Z}_{(t)}^{v_1}, \mathbf{Z}_{(t)}^{v_2}\right) - 1, \tag{20}$$

thus, the original problem of maximizing the fusion score is equivalently transformed into minimizing the matching cost:

$$\min_{\mathbf{X}^{(t)}} \sum_{v_1=1}^{N_t} \sum_{v_2=1}^{N_t} A_{v_1 v_2}^{(t)} \cdot x_{v_1 v_2}^{(t)}, \tag{21}$$

the pairing matrix $\mathbf{X}^{(t)} \in \{0, 1\}^{N_t \times N_t}$ satisfies constraints such as symmetry and unique pairing.

To satisfy the requirement of complete matching for the Hungarian algorithm, if the number of potential representations $N_t$ in the current round is odd, we introduce a dummy node $v_0$ and construct an extended cost matrix $\tilde{\mathbf{A}}^{(t)} \in \mathbb{R}^{N_t' \times N_t'}$, where:

$$N_t' = \begin{cases} N_t, & \text{if } N_t \text{ is even;} \\ N_t + 1, & \text{if } N_t \text{ is odd,} \end{cases} \tag{22}$$

the elements of the extended cost matrix are defined as:

$$\tilde{A}_{v_1 v_2}^{(t)} = \begin{cases} A_{v_1 v_2}^{(t)}, & v_1, v_2 \leq N_t, \\ 0, & v_1 = N_t' \text{ or } v_2 = N_t', v_1 \neq v_2, \\ +\infty, & v_1 = v_2 = N_t'. \end{cases} \tag{23}$$

On this cost matrix, the Hungarian algorithm iteratively searches for a minimum total cost perfect matching through steps such as row and column normalization, zero covering, and auxiliary matrix adjustments. The algorithm ultimately outputs a matching matrix:

$$\tilde{\mathbf{X}}^{(t)} \in \{0, 1\}^{(N_t+1) \times (N_t+1)}, \tag{24}$$

where $\tilde{x}_{v_1 v_2}^{(t)} = 1$ indicates that the $v_1$-th representation is matched with the $v_2$-th representation. The matching matrix is then mapped back to the original representation set, where all pairs involving the virtual node are removed to obtain the actual set of view pairings:

$$\mathcal{P}^{(t)} = \left\{ (v_1, v_2) \mid \tilde{x}_{v_1 v_2}^{(t)} = 1, \quad v_1, v_2 \leq N_t \right\}. \tag{25}$$

### A.2  Complexity Analysis

Consider that the number of views in the $t$-th iteration is $N_t$. The computational complexity of the Hungarian algorithm in this iteration is $T_t = O\left(N_t^3\right)$. As the iterations proceed, the number of views approximately halves each round, i.e., $N_t \approx \frac{V}{2^t}$, where $V$ denotes the total number of initial views.

Therefore, the entire fusion process proceeds for $\log_2 V$ iterations, and the total computational complexity is the sum of the complexities of each iteration:

$$T_{\text{total}} = \sum_{t=1}^{\log_2 V} T_t = \sum_{t=1}^{\log_2 V} O\left(\left(\frac{V}{2^t}\right)^3\right) = O\left(V^3 \sum_{t=1}^{\log_2 V} \frac{1}{8^t}\right). \tag{26}$$

Since the series $\sum_{t=1}^{\infty} \frac{1}{8^t}$ converges to a constant, we have

$$T_{\text{total}} = O(V^3). \tag{27}$$

Table 6: Performance comparison of different methods on various datasets

| Setting | Method | Hdigit | | | 100leaves | | | Fashion | | | MNIST_USPS | | | HandWritten | | |
|---|---|---|---|---|---|---|---|---|---|---|---|---|---|---|---|---|
| | | ACC | NMI | ARI | ACC | NMI | ARI | ACC | NMI | ARI | ACC | NMI | ARI | ACC | NMI | ARI |
| 0.1 | IMC-MCL | 98.27 | 97.05 | 97.61 | 25.38 | 65.70 | 19.30 | 66.11 | 77.00 | 59.98 | 81.72 | 90.35 | 80.67 | 73.33 | 66.28 | 56.94 |
| | MICA | 98.63 | 96.19 | 97.18 | 10.06 | 33.34 | 1.33 | 89.80 | 90.17 | 84.79 | 98.26 | 96.22 | 96.36 | 29.25 | 41.30 | 17.32 |
| | MRL_CAL | 12.63 | 0.73 | 0.28 | 14.75 | 52.75 | 7.92 | 95.12 | 91.50 | 90.09 | 98.38 | 95.76 | 96.44 | 46.65 | 53.79 | 36.52 |
| | ICMVC | 16.56 | 9.10 | 3.82 | 16.80 | 78.38 | 25.52 | 92.31 | 86.82 | 84.90 | 98.12 | 96.48 | 96.06 | 20.37 | 11.38 | 4.82 |
| | APADC | 79.20 | 72.90 | 65.90 | 52.70 | 79.30 | 43.50 | 76.50 | 82.00 | 66.70 | 98.00 | 95.10 | 95.60 | 63.00 | 63.60 | 45.10 |
| | MCAC | 29.57 | 21.85 | 11.92 | 42.60 | 71.13 | 29.45 | 31.97 | 31.20 | 14.34 | 86.34 | 82.63 | 79.84 | 48.87 | 42.78 | 28.89 |
| | DCP | 89.19 | 89.90 | 81.79 | 51.86 | 83.51 | 45.29 | 94.86 | 90.82 | 89.70 | 92.76 | 93.71 | 88.96 | 72.05 | 73.99 | 54.06 |
| | DSIMVC | 91.66 | 90.34 | 88.46 | 32.74 | 66.31 | 21.00 | 91.32 | 86.98 | 83.77 | 98.70 | 96.69 | 97.24 | 74.96 | 73.17 | 64.25 |
| | SURE | 49.01 | 41.76 | 28.61 | 46.34 | 75.55 | 32.28 | 91.56 | 86.97 | 84.14 | 98.45 | 96.30 | 96.62 | 59.57 | 55.84 | 42.89 |
| | COMP | 95.32 | 88.77 | 89.93 | 47.14 | 79.26 | 39.86 | 86.14 | 84.36 | 76.46 | 96.31 | 92.86 | 91.87 | 77.76 | 80.79 | 68.18 |
| | **DFHDC** | **98.96** | **97.22** | **97.72** | **71.16** | **86.93** | **60.93** | **96.42** | **93.13** | **92.52** | **98.75** | **96.79** | **97.27** | **88.77** | **85.79** | **81.68** |
| 0.3 | IMC-MCL | 98.56 | 95.85 | 96.82 | 24.30 | 62.40 | 16.51 | 67.03 | 79.26 | 62.14 | 82.92 | 88.40 | 80.57 | 72.84 | 65.36 | 55.99 |
| | MICA | 69.49 | 82.65 | 70.13 | 9.88 | 34.26 | 2.15 | 88.53 | 89.63 | 83.62 | 69.00 | 79.53 | 66.87 | 31.00 | 46.41 | 22.71 |
| | MRL_CAL | 12.30 | 0.69 | 0.23 | 12.88 | 50.63 | 7.31 | 91.31 | 86.86 | 83.63 | 96.72 | 91.53 | 92.86 | 50.45 | 54.72 | 37.35 |
| | ICMVC | 16.34 | 8.64 | 3.56 | 20.49 | 78.62 | 30.88 | 87.97 | 82.41 | 78.58 | 97.45 | 94.72 | 95.51 | 46.32 | 42.74 | 24.60 |
| | APADC | 84.00 | 81.60 | 74.60 | 38.80 | 70.50 | 27.80 | 76.10 | 80.80 | 65.70 | 96.80 | 92.30 | 93.00 | 70.00 | 72.40 | 50.80 |
| | MCAC | 29.49 | 20.02 | 11.40 | 35.26 | 64.88 | 20.58 | 67.96 | 61.90 | 53.95 | 85.05 | 80.26 | 77.23 | 52.44 | 44.91 | 31.90 |
| | DCP | 94.31 | 93.33 | 89.43 | 51.29 | 80.46 | 45.28 | 91.46 | 86.56 | 83.61 | 95.99 | 94.60 | 92.70 | 70.42 | 77.19 | 55.53 |
| | DSIMVC | 95.30 | 91.38 | 92.30 | 29.78 | 61.91 | 17.53 | 89.5 | 84.63 | 81.06 | 97.01 | 94.74 | 94.63 | 70.42 | 70.21 | 59.41 |
| | SURE | 49.74 | 38.18 | 23.94 | 46.51 | 71.21 | 30.04 | 89.45 | 83.76 | 80.72 | 97.47 | 94.14 | 94.88 | 80.36 | 68.60 | 64.13 |
| | COMP | 93.84 | 85.66 | 86.86 | 38.39 | 72.15 | 28.00 | 76.39 | 79.18 | 69.61 | 92.14 | 90.99 | 88.24 | 72.16 | 74.84 | 52.88 |
| | **DFHDC** | **98.65** | **96.08** | **97.02** | **61.25** | **81.21** | **47.70** | **95.40** | **90.83** | **90.40** | **97.95** | **94.75** | **95.53** | **81.73** | **78.21** | **67.44** |

# B   ADDITIONAL EXPERIMENTAL ANALYSIS

## B.1   EXPERIMENTAL RESULTS UNDER OTHER MISSING RATES

To further evaluate the performance of the proposed method, Table 6 presents experimental results under different missing rate settings, including 0.1 and 0.3. For ease of comparison, the best results are highlighted in bold, and the second-best results are underlined. Experimental results demonstrate that our method exhibits significant advantages in the vast majority of scenarios.

## B.2   ABLATION STUDY

To further verify the effectiveness of each loss component, ablation experiments were conducted on two additional datasets, MNIST_USPS and Fashion. Consistent with the experiments in the main text, the four components $\mathcal{L}_{\text{REC}}$, $\mathcal{L}_{\text{LOC}}$, $\mathcal{L}_{\text{GLO}}$, and $\mathcal{L}_{\text{REF}}$ were evaluated both individually and jointly.

The experimental results (see Appendix Table 7) show that the full-component optimization scheme DFHDC achieves the best clustering accuracy on both datasets, further validating the effectiveness of the proposed dendritic fusion mechanism, hierarchical dual contrastive learning, and fine-tuning completion strategy in complex multi-view clustering tasks.

## B.3   PARAMETER ANALYSIS

To further validate the impact of the loss function hyperparameters on the performance of the proposed method, we conducted parameter sensitivity experiments on four additional datasets: Hdigit, Fashion, MNIST_USPS, and HandWritten. Following the same experimental setup as in the main text, we fixed the reconstruction loss weight $\lambda_1$ at 0.1 and performed pairwise combinations of $\lambda_2$, $\lambda_3$, and $\lambda_4$ within the range $\{0.01, 0.1, 1, 10, 100\}$ for sensitivity analysis. As shown in Fig. 4, experimental results show that when the two critical hyperparameters $\lambda_2$ and $\lambda_3$ are imbalanced, the model performance drops significantly. The best performance is observed when they are maintained at an approximate ratio of $1:10$, which fully validates the effectiveness of our proposed hierarchical dual contrastive learning strategy. The two contrastive objectives complement each other and work collaboratively to optimize the model.

Table 7: Loss ablation experiments on two datasets with the missing rate of 0.5

| | **Loss Components** | | | | **MNIST_USPS** | | | **Fashion** | | |
|---|---|---|---|---|---|---|---|---|---|---|
| | $\mathcal{L}_{\text{REC}}$ | $\mathcal{L}_{\text{LOC}}$ | $\mathcal{L}_{\text{GLO}}$ | $\mathcal{L}_{\text{REF}}$ | ACC | NMI | ARI | ACC | NMI | ARI |
| M-1 | ✓ | | | | 25.45 | 26.23 | 11.50 | 19.13 | 18.03 | 10.77 |
| M-2 | | ✓ | | | 59.43 | 56.48 | 31.06 | 59.59 | 56.93 | 33.46 |
| M-3 | | | ✓ | | 40.08 | 52.64 | 33.56 | 40.45 | 49.64 | 31.70 |
| M-4 | | | | ✓ | 20.26 | 22.18 | 10.06 | 28.54 | 44.59 | 22.33 |
| M-5 | | | ✓ | ✓ | 43.63 | 61.44 | 40.82 | 44.80 | 65.94 | 44.28 |
| M-6 | ✓ | | ✓ | | 33.42 | 43.48 | 23.31 | 41.67 | 48.46 | 30.61 |
| M-7 | ✓ | | | ✓ | 55.10 | 57.77 | 43.91 | 62.26 | 69.61 | 51.85 |
| M-8 | | ✓ | ✓ | | 63.78 | 57.84 | 31.85 | 62.26 | 58.63 | 33.75 |
| M-9 | ✓ | ✓ | | | 68.72 | 66.94 | 44.10 | 60.19 | 55.33 | 33.98 |
| M-10 | | ✓ | | ✓ | 92.40 | 89.47 | 88.32 | 87.25 | 85.65 | 80.74 |
| M-11 | ✓ | | ✓ | ✓ | 39.37 | 58.74 | 38.14 | 58.92 | 73.61 | 53.06 |
| M-12 | ✓ | ✓ | ✓ | | 60.31 | 56.63 | 31.30 | 55.21 | 52.88 | 33.41 |
| M-13 | | ✓ | ✓ | ✓ | 93.62 | 90.71 | 89.96 | 88.27 | 85.35 | 81.20 |
| M-14 | ✓ | ✓ | | ✓ | 96.90 | 92.14 | 93.26 | 89.08 | 85.44 | 82.23 |
| DFHDC | ✓ | ✓ | ✓ | ✓ | **97.06** | **92.49** | **93.59** | **93.21** | **87.35** | **86.28** |

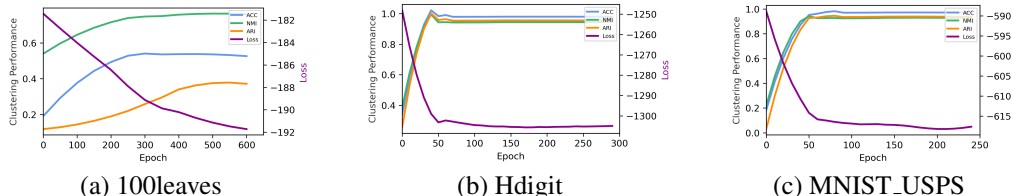

| (a) 100leaves | (b) Hdigit | (c) MNIST_USPS |
|---|---|---|

Figure 3: Clustering metrics (ACC, NMI, ARI) and loss curves over training epochs on the 100leaves, Hdigit, and MNIST_USPS datasets.

## B.4 CONVERGENCE ANALYSIS

To validate the convergence of our proposed method, we conduct experimental analysis on multiple representative datasets, including 100leaves, Hdigit, and MNIST_USPS, by observing the changes in loss values and clustering metrics (ACC, NMI, and ARI) over training epochs. As illustrated in Figure 3 (a), (b), and (c), the curves corresponding to these three datasets are presented. It can be clearly observed that the total loss consistently and significantly decreases as the number of training epochs increases, indicating an effective optimization process. Meanwhile, all clustering performance metrics show a steady upward trend, demonstrating that the model continuously learns more discriminative latent representations during training.

## B.5 VISUALIZATION

As shown in Figure 5, we conduct t-SNE visualization experiments on multiple datasets to further validate the model's ability in capturing the latent feature space. Specifically, Figure 5 (a)–(d) illustrate the clustering effects of common representations on the Hdigit dataset, Figure 5 (e)–(h) show the results for the 100leaves dataset, Figure 5 (i)–(l) present the Fashion dataset, Figure 5 (m)–(p) display the MNIST_USPS dataset, and Figure 5 (q)–(t) show the results for the HandWritten dataset. The visualization results clearly demonstrate that as training progresses, the common representations on all datasets gradually form more distinct cluster structures in the feature space: the inter-class boundaries become increasingly clear with better separation among classes, while intra-class samples exhibit stronger compactness and consistency.

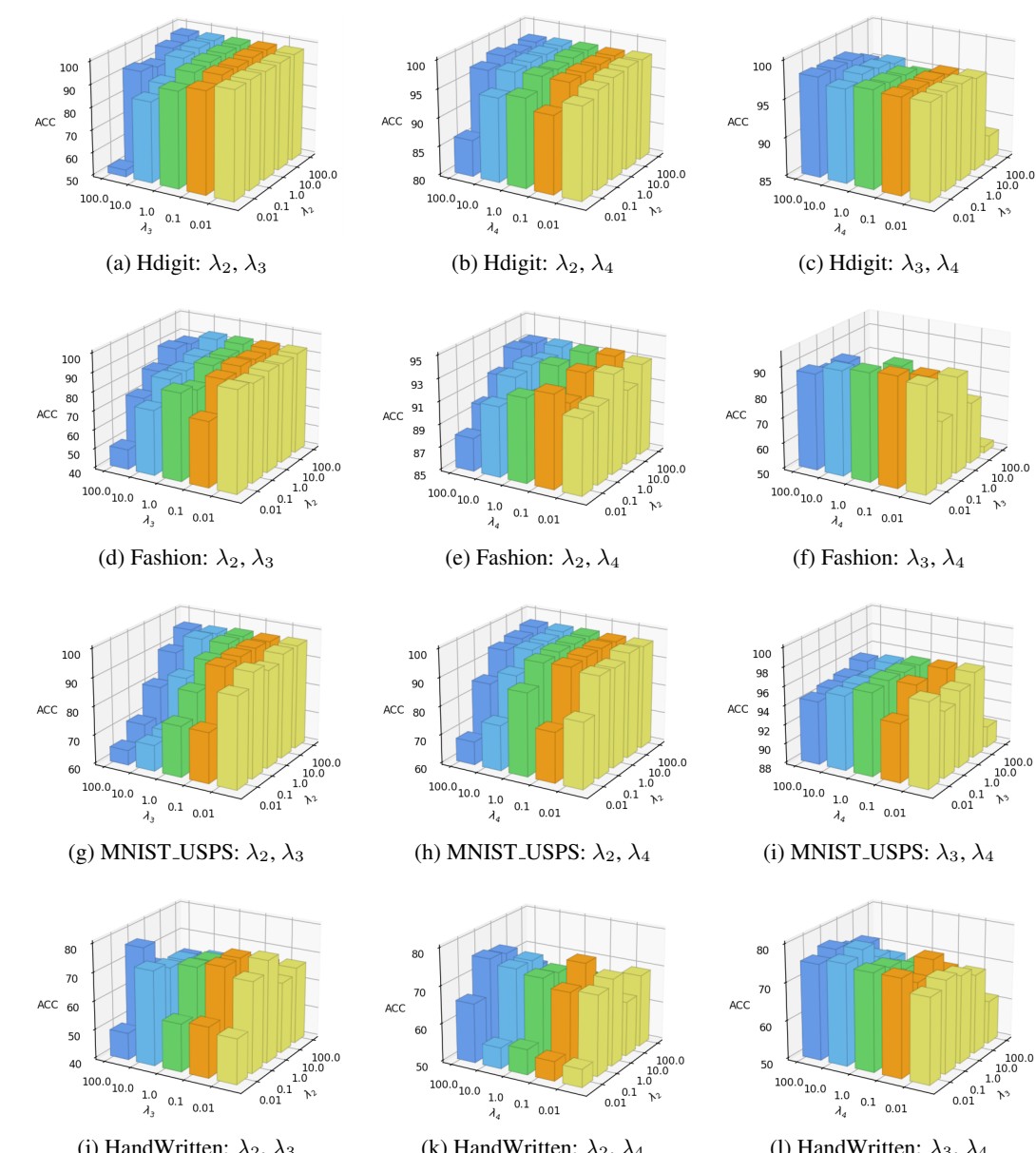

Figure 4: Parameter sensitivity analysis on four datasets (Hdigit, Fashion, MNIST_USPS, Hand-Written) with a missing rate of 0.5 and $\lambda_1 = 0.1$.

## C  USE OF LARGE LANGUAGE MODELS

We only used Large Language Models (LLMs) for language polishing, such as improving the grammar, readability, and clarity of the paper. No part of the research ideas, technical content, experimental design, or results was generated by LLMs.

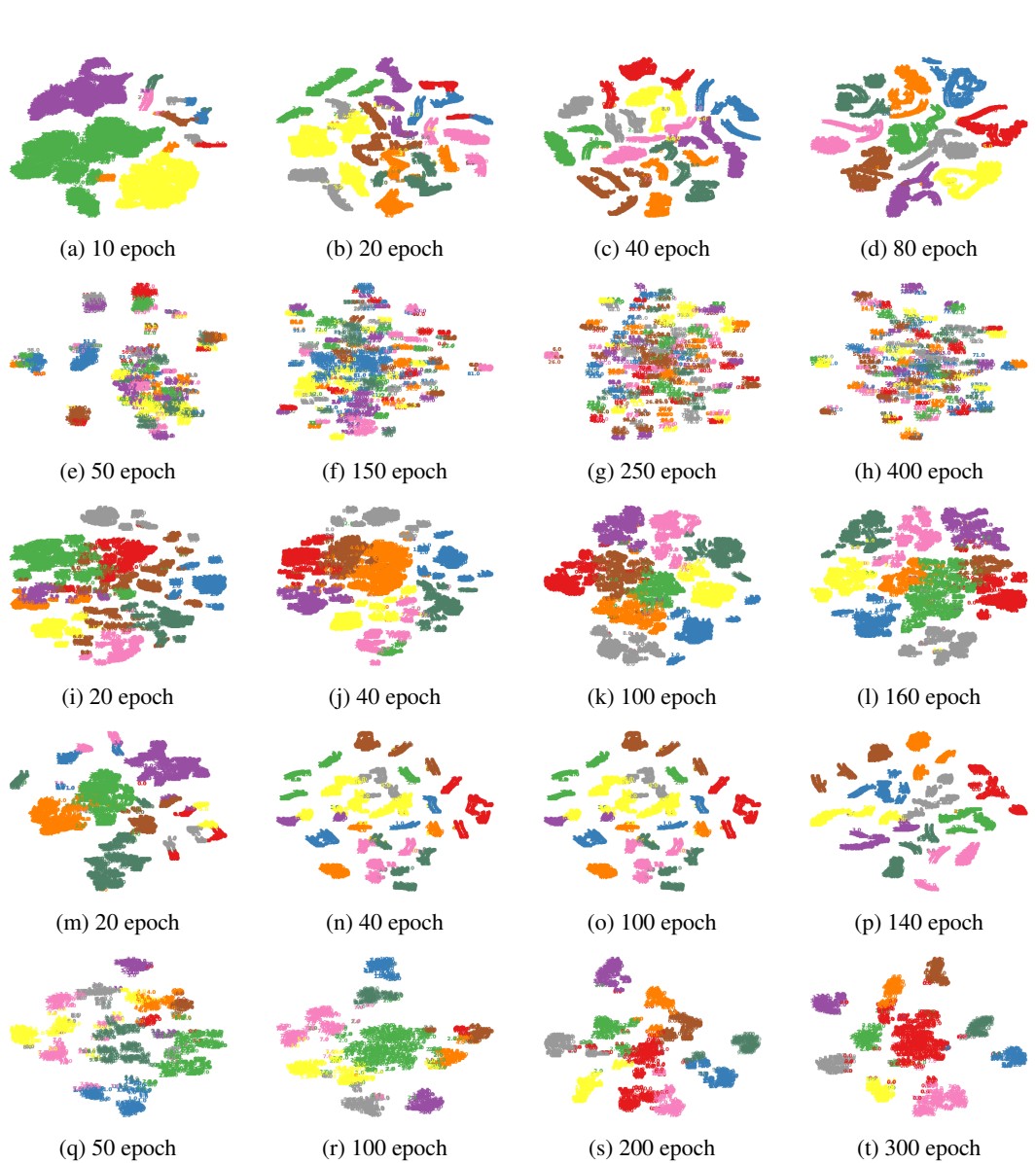

Figure 5: t-SNE visualizations of clustering results on all datasets with increasing training iterations.

