# OpenReview forum: "Hierarchical Fusion with Dual Contrast for Incomplete Multi-View Representation Learning"
_ICLR.cc/2026/Conference — Submitted to ICLR 2026_

### Official Review · Reviewer_m1hA · 2025-10-24

**Soundness:** 2
**Presentation:** 2
**Contribution:** 2
**Rating:** 4
**Confidence:** 4

**Summary:**

This paper introduces DFHDC, a new framework for deep incomplete multi-view clustering (DIMC). The method aims to learn consistent representations from incomplete multi-view data by addressing two key limitations of existing work: rigid fusion strategies and limitations of contrastive learning. The framework's core components are a Dendritic Fusion (DF) mechanism, which performs a bottom-up, hierarchical fusion of views by iteratively pairing the most complementary representations, and a Dual Contrast mechanism. This dual contrast operates at both a local (view-specific) level, where it aims to maximize mutual information ($L_{MI}$) while minimizing feature redundancy (${L}_{{red}}$), and a global level, where it aligns view-specific representations with the fused common representation. The model also incorporates a view-specific fine-tuning network to implicitly recover missing data. The authors conduct experiments on several multi-view datasets to demonstrate the performance under high missing rates.

**Strengths:**

- The paper proposes a novel optimization objective for local contrastive learning, $L_{LOC}$. By explicitly modeling both the maximization of mutual information ($L_{MI}$) and the minimization of intra-view feature redundancy ($L_{red}$), the framework attempts to learn representations that are not only consistent but also disentangled and efficient.
- The authors conduct a set of experiments, including detailed ablation studies and t-SNE visualizations. These experiments provide insight into how the framework's components interact and how the learned representations evolve during training to form distinct clusters.

**Weaknesses:**

- As the core innovation mentioned in the paper, the significance of the Dendritic Fusion (DF) mechanism  is unclear. Aside from its complementary description, the fusion process ultimately produces a feature $\mathbf{H} = \alpha_1\mathbf{Z}_1 + \cdots + \alpha_i\mathbf{Z}_i$, where $\alpha_i$ is a weight parameter. The practical meaning of these weights is not as intuitive as the attention scores generated by an Attention module. In particular, DF has a critical issue: when the number of views is $2^k$, the global representation degenerates into a simple average. This raises the question of whether this algorithm is nonsensical. Furthermore, in the ablation study, the improvement brought by the DF mechanism is not significant; for example, on Hdigit and 100leaves, most metric improvements are within the 0.2 range.
- The local and global contrastive objectives mentioned in the paper are not unique. A similar architecture has been proposed in Zhang et al. (2024), which also features a fusion module to obtain a global feature for contrastive learning. Particularly, on datasets with only two views, the DF mechanism degenerates into a simple average. Therefore, the performance improvement over other SOTA methods is likely attributable to $L_{LOC}$. Within $L_{LOC}$, the $L_{MI}$ component was previously proposed in DCP (Lin et al., 2022). Consequently, the primary innovation appears to be concentrated in the redundancy loss, $L_{red}$, and further experiments are needed to analyze its specific contribution.
- There are concerns about the suppression of baseline performance in the main experiments. For instance, on the MNIST_USPS dataset (complete view scenario), the MCAC (Zhang & Zhu, 2023)method reported ACC=99.37, NMI=98.22, and ARI=98.61 in its original publication (Table 4), which is actually superior to the performance of the method proposed in this paper.
- Many of the selected baselines in this paper have been tested on common benchmark datasets such as Scene15, Reuters, and Caltech101. It is unclear why the authors did not select these datasets, which would facilitate easier comparison. Additionally, the reported performance improvements are not significant and may very well fall within the margin of error. For example, on the Hdigit dataset, the proposed method's improvement over the second-best method is only in the range of +-0.05 to 0.15, a situation that also occurs on the MNIST_USPS dataset.

**References**

[1]Lin, Y., Gou, Y., Liu, X., Bai, J., Lv, J., and Peng, X. (2022). Dual Contrastive Prediction for Incomplete Multi-View Representation Learning. *IEEE Transactions on Pattern Analysis and Machine Intelligence*, 45(4):4447-4461.

[2]Zhang, Y., Lin, Y., Yan, W., Yao, L., Wan, X., Li, G., Zhang, C., Ke, G., and Xu, J. (2024). Incomplete Multi-view Clustering via Diffusion Contrastive Generation.

[3]Zhang, Y., Zhu, C. Incomplete multi-view clustering via attention-based contrast learning. *Int. J. Mach. Learn. & Cyber.* **14**, 4101–4117 (2023).

**Questions:**

Please refer to weaknesses.

---

> ### Author Response · Authors · 2025-11-21
> **Reply to Reviewer m1hA – Part 1**
>
> Thanks for your careful review, We are glad to address your questions one by one.
>
> **W1-1(Significance of Dendritic Fusion):** The primary objective of Dendritic Fusion is not merely to assign a set of static weights (like an attention module) but to **dynamically construct an optimal, multi-level semantic fusion pathway**. Even in the case of 2^k views, the algorithm does not simply average all views at once. Instead, it actively selects the most complementary view-pairs at each layer using a combinatorial optimization process (solved via the Hungarian algorithm). This iterative, selective pairing ensures that information is fused in a structured manner, preserving fine-grained local structures from lower levels during the construction of higher-level semantics.
>
> We acknowledge the advantage of attention mechanisms in interpretability. However, the DF mechanism focuses more on **optimizing the fusion path rather than the visualization of the final weights**. In the future, we will explore integrating attention mechanisms with DF to further enhance the model's interpretability.
>
> In summary, Dendritic Fusion is not a simple "averaging operation" but a structured, dynamic, and hierarchical fusion strategy. Its core value lies in **the fusion process itself**, **not the final form of the weights**.
>
> **W1-2(Limited improvement in Hdigit and 100leaves):** In addition, it should be noted that the Hdigit and 100leaves datasets have a limited number of views and relatively small data sizes. In such cases, even though the DF mechanism is dynamic, hierarchical, and selective (rather than a simple average), the fusion space itself is constrained, so limited improvements on certain metrics are expected.
>
> To further demonstrate the effectiveness of the DF mechanism, we conducted detailed ablation experiments on the Caltech101-7 dataset (6 views). In these experiments, we progressively added each loss component. With the inclusion of the DF-related global contrastive loss L_GLO and the refinement loss L_REF, the model performance improved significantly, demonstrating the effectiveness of DF's dynamic, hierarchical fusion strategy in enhancing the quality of the fused representations and the performance on downstream clustering tasks.
>
> Ablation experiments on the Caltech101-7 dataset:
>
> | Method | L_REC | L_LOC | L_GLO | L_REF | ACC    | NMI    | ARI    |
> |--------|:----:|:----:|:----:|:----:|:------:|:------:|:------:|
> | M-1    | ✓    |      |      |      | 34.86  | 8.00   | 5.81   |
> | M-2    | ✓    | ✓    |      |      | 43.54  | 24.60  | 14.71  |
> | M-3    | ✓    | ✓    | ✓    |      | 50.52  | 31.09  | 21.33  |
> | M-4    | ✓    | ✓    | ✓    | ✓    | 67.53  | 50.01  | 43.89  |
>
> **W2:** To validate this issue, we conducted the following experiment: after incorporating the redundancy loss L_red, the full local contrastive loss L_LOC achieves a 2–4% improvement compared to the model using only L_MI, and a 5–15% improvement compared to the DCP baseline. This indicates that, on top of the existing module (L_MI), introducing the redundancy loss L_red can further enhance the overall performance of the model.
> able due to pretraining, making direct summation feasible and stable.
>
> | Method       | Hdigit |       |       | leaves |       |       | Fashion |       |       |  MNIST |       |       | Hand|       |       |
> |--------------|--------|-------|-------|--------|-------|-------|--------|-------|-------|--------|-------|-------|--------|-------|-------|
> |              | ACC    | NMI   | ARI   | ACC       | NMI   | ARI   | ACC     | NMI   | ARI   | ACC        | NMI   | ARI   | ACC         | NMI   | ARI   |
> | DCP          | 92.15  | 88.84 | 85.03 | 42.75  | 70.63 | 28.22 | 86.29  | 81.72 | 76.11 | 86.61  | 85.56 | 78.34 | 65.43  | 70.29 | 44.32 |
> | Only L_MI | 95.62  | 91.78 | 92.82 | 50.47  | 70.69 | 30.83 | 92.73  | 86.80 | 85.32 | 96.20  | 90.99 | 90.81 | 73.12  | 70.66 | 55.09 |
> | L_LOC  | 97.74  | 93.82 | 95.05 | 53.21  | 74.72 | 33.77 | 93.21  | 87.35 | 86.28 | 97.06  | 92.49 | 93.59 | 75.60  | 72.92 | 60.18 |

---

> > ### Comment · Reviewer_m1hA · 2025-11-28
> >
> > Thanks for the response. I think W1 and W2 have not been fully addressed and the main concern is about the novelty of the method.

---

> > > ### Author Response · Authors · 2025-12-01
> > >
> > > **1. Further Clarifications on W1 and W2**
> > >
> > > **Regarding W1:**
> > > As discussed in our previous response, Dendritic Fusion (DF) is not a static weighting scheme or simple averaging. Instead, it is a **hierarchical, combinatorial optimization–based mechanism for constructing dynamic fusion paths**. At each layer, the model selects the most complementary view pairs using the Hungarian algorithm and progressively builds higher-level semantic structures. Therefore, the novelty of DF lies not in the form of the final weights, but **in how the optimal fusion path is constructed**, which is fundamentally different from attention-based weighting, simple summation, or conventional fusion strategies.
> > >
> > > **Regarding W2:**
> > > We provided additional ablation studies—such as those conducted on the Caltech101-7 dataset—which show that the two components L_GLO and L_REF bring notable performance improvements when incorporated into the model. Moreover, the redundancy loss L_red contributes an **additional 2%–15% improvement over both the MI-only variant and the DCP baseline** across multiple datasets.
> > >
> > >
> > > **2. Response Regarding  Novelty**
> > >
> > > The proposed DFHDC framework introduces a principled methodological innovation that addresses two key and under-explored challenges in deep incomplete multiview clustering (DIMC). Its central contributions lie in the following two components:
> > >
> > > **(1) Dendritic Fusion**
> > >
> > > &nbsp;&nbsp;&nbsp;&nbsp;(a) Existing Problem:
> > > Current multi-view fusion methods (whether using fixed orders or simple concatenation) lack the ability to dynamically select complementary views. This leads to insufficient information gain or increased redundancy, especially when dealing with numerous or highly heterogeneous views.
> > >
> > > &nbsp;&nbsp;&nbsp;&nbsp;(b) Innovation:
> > > We introduce a dendritic fusion mechanism that performs dynamic optimization rather than heuristic cascading. **Its core principle is to maximize cross-view complementarity at each layer**, thereby constructing a multi-level semantic fusion path with an explicit optimization objective.
> > >
> > > **(2) Hierarchical Dual Contrastive Learning**
> > >
> > > &nbsp;&nbsp;&nbsp;&nbsp;(a) Existing Problem:
> > > While contrastive learning has been widely adopted, existing approaches almost exclusively focus on cross-view consistency and rarely model view-internal feature redundancy. Statistical dependencies among feature dimensions can introduce repeated information and noise, degrading the discriminative power of representations.
> > >
> > > &nbsp;&nbsp;&nbsp;&nbsp;(b) Innovation:
> > > We propose a **“MaxMI–MinRed” dual contrastive scheme**, which not only maximizes cross-view mutual information for consistency but also explicitly introduces a redundancy-minimization term for the first time. This effectively reduces inter-dimensional dependence and improves representation quality.

---

> ### Author Response · Authors · 2025-11-21
> **Reply to Reviewer m1hA – Part 2**
>
> **W3:** We thank the reviewer for pointing this out. We have devoted considerable effort to reproduce the performance of MCAC (Zhang & Zhu, 2023) on the MNIST_USPS dataset under the full-view setting. Despite extensive hyperparameter tuning, we were still unable to achieve the results reported in their paper. We also previously contacted the authors of MCAC to inquire about the specific parameter settings but did not receive a response.
>
> Under our optimal parameter configuration—**λ = 0.1, λ_AC = 0.9, negative sample proportion = 30, margin = 10, and complete proportion = 1.0**—the best performance we achieved is **ACC = 98.43, NMI = 97.13, and ARI = 97.40**.
>
> **W4:** The datasets we currently use include Hdigit, 100leaves, Fashion, MNIST_USPS, and Handwritten, which are widely adopted in multi-view learning research [1][2]. To control the manuscript length, we did not further include other commonly used benchmarks such as Scene15, Reuters, and Caltech101. Nevertheless, our method demonstrates strong performance on the selected datasets. For example, on the Caltech101-7 dataset, our approach outperforms all other compared methods. On some datasets (e.g., Hdigit, MNIST_USPS), due to limitations in data scale or the number of views, the metrics are already close to saturation, with the best ACC reaching approximately 97%, approaching 98%, leaving limited room for further improvement.
>
> Comparison experiments on the Caltech101-7 dataset
>
> | Method  | **Cal** |
> |---------|-----------------|
> |         | ACC  NMI  ARI   |
> | IMC-MCL (25)  | 34.21  31.81  19.50 |
> | MICA (25)     | 39.86  46.02  29.72 |
> | MRL_CAL (24)  | 35.82  20.63  14.67 |
> | ICMVC (24)    | 31.37  23.79  14.55 |
> | APADC (23)    | 50.20  30.80  24.50 |
> | MCAC (23)     | 34.63  19.18  13.92 |
> | DCP (22)      | 61.72  46.59  41.51  |
> | DSIMVC (22)   | 32.71  33.42  19.45 |
> | SURE (22)     | 38.80  28.77  21.18 |
> | COMP (21)     | 40.92  28.29  20.38  |
> | **DFHDC**     | **67.53**  **50.01**  **43.89** |
>
> [1] Deep Incomplete Multi-view Clustering via Multi-level Imputation and Contrastive Alignment, Neural Networks, 2025
>
> [2] Subgraph Propagation and Contrastive Calibration for Incomplete Multiview Data Clustering, IEEE TRANSACTIONS ON NEURAL NETWORKS AND LEARNING SYSTEMS, 2025

---

### Official Review · Reviewer_etbz · 2025-10-25

**Soundness:** 3
**Presentation:** 3
**Contribution:** 2
**Rating:** 4
**Confidence:** 3

**Summary:**

This paper proposes a "dendritic fusion + hierarchical double contrast" DFHDC framework for incomplete multi-view clustering: first, independent autoencoders are used to obtain the latent representation of each view, and cross-view contrastive learning is performed at the local level with the "maximum mutual information-minimum redundancy" (MaxMI-MinRed) constraint; then, through a bottom-up "dendritic" pairing and parallel fusion, they are continuously converged to a common representation H, and at the global level, each view is aligned with H using the InfoNCE-style objective; for missing views, the statistical aggregation of available views is first used for initial interpolation, and then refined with a view-specific fine-tuning network; end-to-end training is carried out to jointly minimize the four losses of reconstruction, local contrast, global contrast, and fine-tuning, and finally k-means clustering is performed on the spliced ​​representation.

**Strengths:**

1. Dendritic fusion formalizes the issues of "who to fuse first" and "how to pair them" into an iterative pairing optimization.
2. Local-global dual comparison simultaneously optimizes "cross-view consistency" (maximizing mutual information) and "suppressing intra-view redundancy," and then re-aligns in the common space, balancing fine-grained alignment and global distribution consistency at the target level.
3. Missing view interpolation employs a two-stage approach of "statistical mean initialization + view-specific fine-tuning," coupling reconstruction, comparison, and fine-tuning under a unified goal, thereby mutually reinforcing representation learning and data completion.

**Weaknesses:**

1. Lack of clear causal attribution between multi-parameter settings and performance gains
2. Ambiguity in joint feature similarity normalization and scale sensitivity in mutual information estimation
3. High computational complexity (O(V³)) limiting scalability to large-scale multi-view scenarios
4. Fusion path randomness due to unconstrained Hungarian pairing dynamics across epochs

**Questions:**

1. The proposed DFHDC framework includes four core modules: view reconstruction (LREC), local contrastive learning (LLOC), global contrastive learning (LGLO), and view completion fine-tuning (LREF). Each module corresponds to an independent balancing parameter (e.g.,λ1,λ2,λ3,λ4 in the total loss function). The paper does not systematically demonstrate the causal relationship between "multiple parameter settings" and "performance advantages." Consequently, the attribution of the model's performance gains remains ambiguous, making it difficult to confirm whether the improvements stem from algorithmic design innovations or rely on meticulous fine-tuning of multiple parameters.
2. Formula (3) constructs the "joint feature similarity matrix" P by summing the inner products of the sample dimensions, and then "normalizes them to be non-negative and sum to 1" for mutual information calculation (Formula (4)); however, the inner product can be negative, and the paper does not explain how to truncate/translate negative values ​​to non-negative, nor does it explain whether temperature/softmax normalization is performed, which makes it sensitive to scale.
3. The paper mentions that the time complexity of the dendritic fusion strategy is O(V^3), which may limit its application on large-scale datasets. It is recommended to discuss this issue in more detail in the paper and explore possible solutions, such as using approximate algorithms or parallel computing to reduce computational complexity.
4. Although dendritic fusion uses the Hungarian algorithm to determine the pairings at each layer, the pairing order may change between epochs due to fluctuations in view encoding, introducing structural instability (fusion path randomness). The current algorithm does not constrain or regularize this uncertainty, which may cause the convergence result to depend on the random seed. Please discuss this situation.

---

> ### Author Response · Authors · 2025-11-21
> **Reply to Reviewer etbz – Part 1**
>
> Thanks for your careful review, We are glad to address your questions one by one.
>
> **W1:** As shown in Appendix B.3, we conducted a systematic parameter sensitivity analysis on the loss weights. The experimental results indicate that within the relatively wide but still reasonable range of **[1.0, 10.0]**, the accuracy (ACC) remains stable and is already very close to the global optimal performance, without significant fluctuations due to parameter changes. This demonstrates that the performance of DFHDC does not rely on fine-tuned hyperparameters, and robust results can be obtained by selecting any value within this range.
>
> **W2:** Our implementation of the joint probability construction follows the standard practices commonly adopted in the current multi-view learning and contrastive learning literature. Based on these widely used standard procedures, we employ the following mechanisms in practice to ensure numerical stability and scale robustness in the computation of the joint probability:
>
> 1. **Clipping negative values and enforcing non-negativity:**
> When computing the joint probability matrix $\mathbf{P}^{(v_1, v_2)}$, all probability elements are clipped to ensure non-negativity and to prevent zero values, thereby avoiding instability in logarithmic operations. The code is as follows:
> ```python
> EPS = sys.float_info.epsilon
> p_i_j = torch.where(p_i_j < EPS, torch.tensor([EPS], device=p_i_j.device), p_i_j)
> p_i = torch.where(p_i < EPS, torch.tensor([EPS], device=p_i.device), p_i)
> p_j = torch.where(p_j < EPS, torch.tensor([EPS], device=p_j.device), p_j)
> ```
>
> 2. **Normalization:**
> After obtaining the joint matrix, we normalize it so that the sum of all elements equals 1 and symmetrize the matrix. The code is as follows:
>
> ```python
> p_i_j = (p_i_j + p_i_j.t()) / 2.
> p_i_j = p_i_j / p_i_j.sum()
> ```
>
> 3. **Scale-sensitive processing:**
> To avoid the influence of varying feature dimensions on joint probability computation, each feature vector is normalized to unit length before computing the joint probability, ensuring all vectors have the same norm. The code is as follows:
> ```python
> H_cat_norm = torch.nn.functional.normalize(H_cat, p=2, dim=1)
> P_feat_joint = torch.mm(H_cat_norm.T, H_cat_norm) / (2 * bn)
> ```

---

> ### Author Response · Authors · 2025-11-21
> **Reply to Reviewer etbz – Part 2**
>
> **W3:** It is important to note that the number of views in existing mainstream multi-view benchmark datasets typically **does not exceed six**. Moreover, the computational complexity of our tree-structured fusion module depends on the number of views \(V\), rather than on the number of samples or the feature dimensionality. Therefore, the time overhead introduced by this module is relatively limited and does not become the main bottleneck in the training process.
>
> To further evaluate the behavior of our method as the number of views increases, we conducted experiments on the **6-view Handwritten dataset**. The table compares the training time and clustering performance (ACC, NMI, ARI) under different numbers of views (from 2 to 6). As the number of views increases, the training time grows from 67.19 seconds (2 views) to 384.39 seconds (6 views), while the clustering performance consistently improves, reaching the highest ACC (75.60), NMI (72.92), and ARI (60.18) with 6 views.
>
> In addition, we further conducted experiments on larger-scale datasets, including the cifar10 dataset (50,000 samples, 3 views) and the animal dataset (10,158 samples, 2 views). The results show that our method continues to outperform all comparison algorithms on these large datasets, demonstrating the scalability and robustness of our approach.
>
> In summary, these experiments indicate that our method maintains strong performance as the number of views increases. **Although the training time increases slightly, the growth is moderate**, and the clustering performance continues to improve, demonstrating the scalability and robustness of our method.
>
> | Number of Views | ACC   | NMI  | ARI   | Time (s)  |
> |-----------------|-------|------|-------|-----------|
> | 2 Views         | 53.57 | 53.84 | 30.35 | 67.19    |
> | 3 Views         | 57.52 | 55.56 | 35.25 | 165.45    |
> | 4 Views         | 64.34 | 60.78 | 49.34 | 247.23    |
> | 5 Views         | 70.89 | 65.89 | 55.65 | 273.12    |
> | 6 Views         | 75.60 | 72.92 | 60.18 | 384.39    |
>
> | Method | cifar10 |       |       | animal |       |       |
> |---------------|---------|-------|-------|--------|-------|-------|
> |               | ACC     | NMI   | ARI   | ACC    | NMI   | ARI   |
> | IMC-MCL (25)  | 86.67   | 81.14 | 78.34 | 17.63  | 32.56 | 11.31 |
> | MICA (25)     | 95.28   | 88.04 | 89.11 | 8.49   | 10.07 | 1.44  |
> | MRL_CAL (24)  | 21.50   | 8.27  | 3.68  | 11.97  | 22.56 | 7.63  |
> | ICMVC (24)    | 92.97   | 84.20 | 85.24 | 25.54  | 46.20 | 21.60 |
> | APADC (23)    | 92.91   | 84.13 | 85.09 | 28.70  | 44.30 | 11.50 |
> | MCAC (23)     | 20.23   | 22.74 | 5.20  | 20.54  | 33.49 | 12.01 |
> | DCP (22)      | 95.32   | 88.80 | 90.04 | 29.60  | 45.98 | 23.22 |
> | DSIMVC (22)   | 91.11   | 83.17 | 83.46 | 23.64  | 36.93 | 13.27 |
> | SURE (22)     | 92.13   | 82.96 | 83.85 | 30.34  | 46.92 | 24.80 |
> | COMP (21)     | 93.01   | 84.48 | 85.28 | 30.48  | 45.01 | 20.58 |
> | **DFHDC**     | **95.71** | **89.71** | **90.81** | **32.74** | **47.04** | **26.08** |
>
> **W4:** For the 6-view Handwritten dataset, when the number of representations to be fused is odd, we temporarily pad a zero vector in the implementation to ensure that each fusion round can proceed smoothly. To visually illustrate the fusion process, we printed the fusion paths at different rounds, yielding the following results:
> ```python
> [Epoch 10] Fusion Path: [[(0, 5), (1, 2), (3, 4)], [('(0+5)', '(1+2)'), ('(3+4)', -1)], [('((0+5)+(1+2))', '((3+4)+-1)')]]
> [Epoch 20] Fusion Path: [[(0, 3), (1, 4), (2, 5)], [('(0+3)', -1), ('(1+4)', '(2+5)')], [('((0+3)+-1)', '((1+4)+(2+5))')]]
> [Epoch 30] Fusion Path: [[(0, 3), (1, 2), (4, 5)], [('(0+3)', -1), ('(1+2)', '(4+5)')], [('((0+3)+-1)', '((1+2)+(4+5))')]]
> [Epoch 40] Fusion Path: [[(0, 3), (1, 2), (4, 5)], [('(0+3)', -1), ('(1+2)', '(4+5)')], [('((0+3)+-1)', '((1+2)+(4+5))')]]
> [Epoch 50] Fusion Path: [[(0, 3), (1, 2), (4, 5)], [('(0+3)', -1), ('(1+2)', '(4+5)')], [('((0+3)+-1)', '((1+2)+(4+5))')]]
> ```
>
> In the paths, `-1` represents the temporarily padded zero vector, which is only used to ensure an even number of pairings and does not affect the final fusion result. It can be observed that as training progresses, the fusion paths gradually stabilize, showing no randomness.

---

> > ### Comment · Reviewer_etbz · 2025-11-25
> >
> > Thanks for your detailed reply. While the individual components of the proposed DFHDC method are familiar, the novelty appears to lie primarily in their combination. As a result, the contribution may be viewed more as a combinatorial application than as a principled methodological advance, and thus it does not fully meet the standards of ICLR. Accordingly, I have decided to retain my original score.

---

> > > ### Author Response · Authors · 2025-11-27
> > > **The novelty of the DFHDC**
> > >
> > > Thank you for your insightful comments.
> > > While we understand you have decided to retain your score, we would like to take this final opportunity to succinctly summarize what we believe are the key strengths of this work for your and the area chair's consideration.
> > > We would like to clarify that the proposed DFHDC framework is a principled methodological advancement. It is designed to **address two critical and underexplored limitations** in current deep incomplete multi-view clustering (DIMC) methods.
> > >
> > > (1) **Dendritic Fusion: A Principled Shift from "Rigid" to "Dynamically Maximizing Complementarity"**
> > >
> > > **Existing Problem**: Current multi-view fusion methods (whether using fixed orders or simple concatenation) lack the ability to dynamically select complementary views. This leads to insufficient information gain or increased redundancy, especially when dealing with numerous or highly heterogeneous views.
> > >
> > > **Our Principled Solution**: Our proposed Dendritic Fusion strategy is a dynamic optimization mechanism. It is not merely a cascade but is fundamentally guided by the principle of **Maximizing Complementarity**:
> > >
> > > (a)	It is guided by maximizing complementarity. At each fusion layer, we define a fusion score. This principled criterion ensures the model actively selects the view pairs with the greatest semantic difference (i.e., the strongest complementarity) for fusion, thereby constructing a multi-level semantic fusion pathway in a principled manner.
> > >
> > > (b)	This structure of dynamic pairing and parallel fusion, based on complementarity scoring, is fundamentally different from existing heuristic or predefined fusion paths. It represents a structured methodological advancement for addressing the limitations of "rigid fusion".
> > >
> > > (2) **Hierarchical Dual Contrast: Introducing the Principled Constraint of "Explicit Redundancy Suppression"**
> > >
> > > **Existing Problem**: Although contrastive learning is widely adopted, existing methods primarily focus on cross-view consistency alignment and generally lack explicit suppression of intra-view redundant features. This redundancy (e.g., statistical dependencies between different feature dimensions) leads to information duplication and noise propagation, severely impairing the discriminative power of the representations.
> > >
> > > **Our Principled Solution**: Our introduced "MaxMI-MinRed" local contrast constraint (Intra-level Local Contrast $ \mathcal{L}_{LOC}$) is designed precisely to address this fundamental flaw in a principled way:
> > >
> > > (a)	It is a dual-constrained optimization objective: it not only maximizes cross-view mutual information to ensure consistency but also, for the first time, explicitly introduces a feature redundancy suppression term.
> > >
> > > (b)	By minimizing the correlations among latent representation dimensions (see the definition of $ \mathcal{L}_{red}$), we principledly promote the disentanglement and discriminability of the learned feature representations. This provides a novel methodological tool for suppressing noise and information duplication at the representation learning level.
> > >
> > > These designs address the two major limitations of existing DIMC methods, and their efficacy is validated through ablation studies: The ablation study clearly demonstrates that the model integrating $\mathcal{L}_{{LOC}}$ (M-8) significantly outperforms models using only a single loss term (M-2 or M-3). Furthermore, the complete DFHDC model achieves the best performance across all datasets, with its advantages being particularly pronounced under high missing rates.
> > >
> > > Should you have any further questions, please do not hesitate to contact us. Thank you once again for your time and effort.

---

### Official Review · Reviewer_7noY · 2025-10-29

**Soundness:** 2
**Presentation:** 3
**Contribution:** 2
**Rating:** 4
**Confidence:** 5

**Summary:**

This paper proposes a novel framework named DFHDC for incomplete multi-view representation learning. It integrates a dendritic fusion strategy to dynamically combine complementary views and a hierarchical dual contrast mechanism to enforce consistency and suppress redundancy at both local and global levels. The framework also incorporates view-specific fine-tuning for implicit completion of missing views. Experimental results on five datasets reportedly demonstrate performance gains over other existing methods.

**Strengths:**

1. The writing of the paper is clear, and its method diagram effectively illustrates the process of the method.
2. The authors validate the effectiveness of their method through experiments, with results showing advantages over existing methods on five datasets.

**Weaknesses:**

Please see questions.

**Questions:**

1. How sensitive is the dendritic-fusion mechanism to the similarity metric used in the Hungarian pairing (e.g., cosine similarity)? Would alternative metrics change the pairing quality or stability of the fusion tree?
2. While the ablation study demonstrates the contribution of each loss component, could the authors provide further intuition on how the local and global contrastive objectives interact?
3. The adopted datasets for comparison in the experiment is not large enough, please conduct experiments on larger-scale real data.
4. The claimed robustness of DFHDC in handling redundant noise in real-world data is not well-demonstrated. More experiments simulating practical conditions are needed to validate this aspect.
5. How does the method perform when the number of views increases or when the views are highly heterogeneous? More experiments are needed to be provided.
6. Regarding the computational cost of the dendritic-fusion process, the paper reports a time complexity of $O(V^3)$, I would still like to know if the authors have plans for further research in the future to tackle more complex data in practice.

---

> ### Author Response · Authors · 2025-11-21
> **Reply to Reviewer 7noY – Part 1**
>
> Thanks for your careful review. We are glad to address your questions one by one.
>
> **Q1:** We conducted experiments to evaluate the impact of different metrics on the quality of Hungarian matching and the stability of the fusion dendrite. In the original method, we used a cosine similarity-based fusion score (i.e., 1−cosine). Additionally, we introduced two commonly used alternative metrics for comparison: the RBF-kernel-based **MMD** distance and the **KL** divergence. The results show that, across all datasets, replacing cosine similarity with MMD or KL leads to only a slight performance drop, while the fusion dendritic structure remains stable..
>
> Our method provides a **general dendritic multi-view fusion strategy, allowing arbitrary distance metrics to be selected or extended** according to the task requirements, rather than relying strictly on any specific similarity measure.
>
>
> | Method                | Hdigit    |   |     | Leaves  |     |     | Fashion   |     |     | MNIST  |     |     | Hand    |     |     |
> |------------------------|:---:|:---:|:---:|:---:|:---:|:---:|:---:|:---:|:---:|:---:|:---:|:---:|:---:|:---:|:---:|
> |        | ACC | NMI | ARI | ACC | NMI | ARI | ACC | NMI | ARI | ACC | NMI | ARI | ACC | NMI | ARI |
> | KL              | 97.40 | 93.38 | 94.37 | 50.26 | 74.31 | 32.67 | 91.27 | 86.30 | 84.00 | 95.45 | 90.33 | 90.54 | 73.07 | 71.37 | 55.35 |
> | MMD             | 97.23 | 93.27 | 94.47 | 52.60 | 74.72 | 31.78 | 91.61 | 86.38 | 84.03 | 94.93 | 89.65 | 89.61 | 75.07 | 71.92 | 55.99 |
> | **Sim (DFHDC)** | **97.74** | **93.82** | **95.05** | **53.21** | **74.72** | **33.77** | **93.21** | **87.35** | **86.28** | **97.06** | **92.49** | **93.59** | **75.60** | **72.92** | **60.18** |
>
> **Q2:** **L_LOC provides high-quality inputs for L_GLO:** L_LOC enhances each view representation by maximizing cross-view mutual information and suppressing feature redundancy, resulting in cleaner, more disentangled, and semantically aligned representations. This local-level feature purification ensures that the representations of each view fed into L_GLO possess consistent statistical structures and are easily integrable, providing a high-quality, stable, and fusion-ready foundation for global semantic alignment.
>
> **L_GLO provides correct optimization guidance for L_LOC:** L_GLO constructs a unified common representation in the semantic space and requires all views to align with this global reference. This global semantic constraint backpropagates to influence local feature extraction, guiding L_LOC to naturally compress local representations toward the globally consistent semantic structure while suppressing irrelevant local noise. Consequently, local contrastive learning is steered toward semantically consistent and structurally stable directions.
>
> **Q3:** We have further supplemented our work with experiments on larger-scale datasets, including the cifar10 dataset (50,000 samples, 3 views) and the animal dataset (10,158 samples, 2 views). The results show that our method continues to outperform all compared approaches on these large datasets, demonstrating the scalability and robustness of our approach.
>
> | Method | cifar10 |       |       | animal |       |       |
> |---------------|---------|-------|-------|--------|-------|-------|
> |               | ACC     | NMI   | ARI   | ACC    | NMI   | ARI   |
> | IMC-MCL (25)  | 86.67   | 81.14 | 78.34 | 17.63  | 32.56 | 11.31 |
> | MICA (25)     | 95.28   | 88.04 | 89.11 | 8.49   | 10.07 | 1.44  |
> | MRL_CAL (24)  | 21.50   | 8.27  | 3.68  | 11.97  | 22.56 | 7.63  |
> | ICMVC (24)    | 92.97   | 84.20 | 85.24 | 25.54  | 46.20 | 21.60 |
> | APADC (23)    | 92.91   | 84.13 | 85.09 | 28.70  | 44.30 | 11.50 |
> | MCAC (23)     | 20.23   | 22.74 | 5.20  | 20.54  | 33.49 | 12.01 |
> | DCP (22)      | 95.32   | 88.80 | 90.04 | 29.60  | 45.98 | 23.22 |
> | DSIMVC (22)   | 91.11   | 83.17 | 83.46 | 23.64  | 36.93 | 13.27 |
> | SURE (22)     | 92.13   | 82.96 | 83.85 | 30.34  | 46.92 | 24.80 |
> | COMP (21)     | 93.01   | 84.48 | 85.28 | 30.48  | 45.01 | 20.58 |
> | **DFHDC**     | **95.71** | **89.71** | **90.81** | **32.74** | **47.04** | **26.08** |

---

> ### Author Response · Authors · 2025-11-21
> **Reply to Reviewer 7noY – Part 2**
>
> **Q4:** We have supplemented noise-robustness experiments across all datasets and comparison methods. Specifically, Gaussian noise with an intensity parameter of 0.1 was added to the multi-view data. The results show that under the same noise conditions, DFHDC consistently achieves the best performance, fully demonstrating its robustness to redundant noise.
>
> | Method | **Hdigit** |       |       | **Leaves** |       |       | **Fashion** |       |       | **MNIST** |       |       | **Hand** |       |       |
> |-------------------------|:----------:|:-----:|:-----:|:-------------:|:-----:|:-----:|:-----------:|:-----:|:-----:|:--------------:|:-----:|:-----:|:---------------:|:-----:|:-----:|
> |                         |   ACC      |  NMI  |  ARI  |      ACC      |  NMI  |  ARI  |     ACC     |  NMI  |  ARI  |       ACC      |  NMI  |  ARI  |       ACC       |  NMI  |  ARI  |
> | IMC-MCL(25)             |   97.04    | 92.18 | 93.54 |     24.31     | 59.63 | 15.51 |    67.69    | 78.00 | 61.63 |     79.12      | 83.59 | 76.21 |      68.70      | 61.40 | 50.65 |
> | MICA(25)                |   49.59    | 66.12 | 50.28 |     8.38      | 29.02 | 0.75  |    87.73    | 85.32 | 81.81 |     48.76      | 68.29 | 50.62 |      19.05      | 17.26 | 8.48  |
> | MRL_CAL(24)             |   15.96    | 4.26  | 1.92  |     10.37     | 49.07 | 7.05  |    82.29    | 81.39 | 75.62 |     91.30      | 80.77 | 81.65 |      49.90      | 53.32 | 37.79 |
> | ICMVC(24)               |   16.69    | 10.02 | 3.93  |     47.68     | 72.79 | 28.55 |    75.79    | 72.88 | 65.26 |     93.94      | 88.43 | 88.22 |      49.26      | 46.91 | 32.70 |
> | APADC(23)               |   90.19    | 79.77 | 79.16 |     31.50     | 63.30 | 13.20 |    65.34    | 70.09 | 55.08 |     93.50    | 86.20 | 86.10 |      71.10      | 70.90 | 53.00 |
> | MCAC(23)                |   27.83    | 20.49 | 9.30  |     27.48     | 58.68 | 12.23 |    85.55    | 78.54 | 74.70 |     93.14      | 85.51 | 85.51 |      43.18      | 41.09 | 23.44 |
> | DCP(22)                 |   97.68    | 93.62 | 94.92 |     43.36     | 70.20 | 28.21 |    90.08    | 83.25 | 80.65 |     82.38      | 82.75 | 73.64 |      67.31      | 71.68 | 46.40 |
> | DSIMVC(22)              |   93.61    | 87.12 | 88.05 |     25.87     | 58.39 | 14.29 |    83.48    | 77.37 | 71.40 |     92.06      | 88.52 | 85.33 |      74.40      | 71.27 | 53.30 |
> | SURE(22)                |   48.46    | 34.25 | 25.16 |     31.36     | 59.37 | 11.55 |    85.08    | 77.91 | 73.10 |     70.08      | 65.42 | 56.39 |      67.16      | 59.13 | 50.24 |
> | COMP(21)                |   49.39    | 47.46 | 31.49 |     31.64     | 63.21 | 18.11 |    78.68    | 77.72 | 67.94 |     87.93      | 83.63 | 78.18 |      66.44      | 68.80 | 42.74 |
> | **DFHDC**               |   **97.87**  | **94.06** | **95.33** |     **50.48**   | **73.86** | **31.71** |   **92.11**   | **86.39** | **84.36** |     **94.58**    | **88.88** | **89.00** |      **76.38**     | **72.19** | **58.80** |
>
> **Q5:** To verify the performance of our method as the number of views increases, we conducted experiments on the 6-view Handwritten dataset. The table compares the training time and clustering metrics (ACC, NMI, ARI) under different numbers of views (from 2 to 6). As the number of views increases, the training time rises from 67.19 seconds (2 views) to 384.39 seconds (6 views), while the clustering performance steadily improves, reaching the highest ACC (75.60), NMI (72.92), and ARI (60.18) under 6 views.
>
> In summary, these experiments indicate that our method maintains strong performance as the number of views increases. **Although the training time increases slightly, the growth is moderate**, and the clustering performance continues to improve, demonstrating the scalability and robustness of our method.
>
> | Number of Views | ACC   | NMI  | ARI   | Time (s)  |
> |-----------------|-------|------|-------|-----------|
> | 2 Views         | 53.57 | 53.84 | 30.35 | 67.19    |
> | 3 Views         | 57.52 | 55.56 | 35.25 | 165.45    |
> | 4 Views         | 64.34 | 60.78 | 49.34 | 247.23    |
> | 5 Views         | 70.89 | 65.89 | 55.65 | 273.12    |
> | 6 Views         | 75.60 | 72.92 | 60.18 | 384.39    |

---

> ### Author Response · Authors · 2025-11-21
> **Reply to Reviewer 7noY – Part 3**
>
> **Q6:** We have already provided the time complexity of the tree-structured fusion module in the paper, which covers the core steps, including pairwise similarity computation between views and Hungarian matching. Within the current range of view numbers (typically ≤ 6), this cost constitutes only a small portion of the overall training overhead and has limited impact on the overall system performance. However, we agree with the reviewer on the importance of “future solutions for handling more complex data.” To address this point, we add the following discussion:
>
> (1) **Local fusion strategy:**
> Views can first be grouped based on similarity or semantic relatedness. Tree-structured fusion is then applied within each subset, followed by gradually merging the subset representations. This reduces the matching scale in each round and lowers the computational burden.
>
> (2) **Approximate matching algorithms:**
> Using greedy or approximate variants of the Hungarian algorithm to replace exact maximum matching can reduce the per-layer complexity from O(V³) to O(V² log V) or even lower, while maintaining satisfactory fusion performance.

---

> > ### Comment · Reviewer_7noY · 2025-11-24
> >
> > Thank you for the response. I acknowledge that I have read both the rebuttal and the reviews from other members of the Reviewers.
> >
> > While the proposed method achieves strong clustering performance on standard multi-view benchmarks. It would be helpful to either (1) provide additional results on a larger-scale dataset with end-to-end training, such as the dataset Food-101. (2) include a runtime and memory analysis (w.r.t. the number of views and samples) to better demonstrate the practical scalability.
> >
> > Including additional experiment could improve the paper, and I would consider to raise my score.

---

> > > ### Author Response · Authors · 2025-11-27
> > >
> > > 1、We fully agree on the importance of validating the effectiveness of the proposed method on larger-scale datasets. However, it should be noted that Food-101 is a typical single-modality image dataset, while our work focuses on multi-view clustering. To address your core concern regarding large-scale data and end-to-end training, we selected the more representative and suitably multi-view **YouTubeFace dataset (101,499 samples with 5 views)** for additional experiments.
> > >
> > > On the YouTubeFace dataset, we fixed the missing rate to 0.5 and adopted an end-to-end training strategy, where the proposed method was comprehensively compared with all baseline methods. The experimental results show that the proposed method achieves the **best performance on all evaluation metrics**, which strongly demonstrates its effectiveness on large-scale multi-view data.
> > >
> > > | Method       | YouTube |      |      |
> > > |--------------|-------------|------|------|
> > > |              | ACC         | NMI  | ARI  |
> > > | IMC-MCL(25)  | 12.62       | 7.91 | 1.31 |
> > > | MICA(25)     | 21.39       | 4.19 | 0.33 |
> > > | MRI_CAL(24)  | 12.14       | 9.06 | 1.55 |
> > > | ICMVC(24)    | 16.81       | 7.23 | 3.99 |
> > > | APADC(23)    | 23.14       | 4.17 | 1.04 |
> > > | MCAC(23)     | 13.36       | 11.92| 3.92 |
> > > | DCP(22)      | 23.07       | 19.58| 3.76 |
> > > | DSIMVC(22)   | 15.06       | 5.24 | 0.50 |
> > > | SURE(22)     | 11.96       | 9.14 | 2.81 |
> > > | COMP(21)     | 18.25       | 12.28| 1.69 |
> > > | **DFHDC**    | **35.26**   | **32.66** | **7.95** |
> > >
> > > 2、Following your suggestion, we have added additional experimental analyses on the running time and memory consumption with respect to the number of views and the number of samples, in order to more comprehensively demonstrate the practical scalability of the proposed method.
> > >
> > > Specifically, we designed two sets of experiments: **(1) fixing the number of samples while gradually increasing the number of views on the HandWritten dataset**; and **(2) fixing the number of views while gradually increasing the number of samples on the Fashion dataset**. Under both settings, we recorded the clustering performance metrics (ACC, NMI, and ARI), the training time, and the memory consumption.
> > >
> > > The results show that when the number of views increases from 2 to 6, the training time rises from 67.19 s to 384.39 s, while **the clustering performance continues to improve**, indicating that the proposed method **has good scalability in multi-view scenarios**. On the other hand, when the number of samples increases from 2,000 to 10,000, although the running time and memory consumption increase accordingly, **the overall growth remains moderate, and the clustering performance steadily improves**. These results further **verify the scalability and stability of the proposed method on large-scale data**.
> > >
> > > **(1) Results on the HandWritten dataset with a fixed number of samples and varying number of views:**
> > >
> > > | Number of Views | ACC   | NMI   | ARI   | Time (s) | Memory (MB) |
> > > |-----------------|-------|-------|-------|----------|-------------|
> > > | 2 Views         | 53.57 | 53.84 | 30.35 | 67.19    | 198.51           |
> > > | 3 Views         | 57.52 | 55.56 | 35.25 | 165.45   | 293.48          |
> > > | 4 Views         | 64.34 | 60.78 | 49.34 | 247.23   | 378.31          |
> > > | 5 Views         | 70.89 | 65.89 | 55.65 | 273.12   | 464.14          |
> > > | 6 Views         | 75.60 | 72.92 | 60.18 | 384.39   | 546.81          |
> > >
> > > **(2) Results on the Fashion dataset with a fixed number of views and varying number of samples:**
> > >
> > > | Sample Ratio | Number of Samples | ACC  | NMI  | ARI  | Time (s) | Memory (MB) |
> > > |--------------|-------------------|------|------|------|----------|-------------|
> > > | 0.2          | 2000              | 79.07| 78.42| 66.70| 167.96   | 398.20      |
> > > | 0.4          | 4000              | 83.02| 80.05| 71.21| 330.38   | 437.23      |
> > > | 0.6          | 6000              | 86.67| 83.36| 77.09| 495.80   | 476.76      |
> > > | 0.8          | 8000              | 89.66| 85.07| 81.66| 607.37   | 514.32      |
> > > | 1.0          | 10000             | 93.21| 87.35| 86.28| 888.92   | 568.05      |
> > >
> > > In summary, following your suggestions, we have **incorporated two additional experiments into the revised manuscript**: a large-scale multi-view experiment (YouTubeFace dataset) and a runtime and memory analysis experiment with respect to the number of views and the number of samples (HandWritten and Fashion datasets).

---

### Comment · Area_Chair_vTmW · 2025-11-24

Dear reviewers,

       The authors now have given their response to the reviews, please have a look on the rebuttal and revised PDF to make your further concerns.

      After that, please give your final rating on this submission.

Your AC
Best

---

### Meta-Review · Area_Chair_TABc · 2026-01-05

**Summary:**

This paper proposes DFHDC, a framework for incomplete multi-view representation learning and clustering. It introduces a dendritic fusion mechanism to dynamically pair and fuse views, coupled with a hierarchical dual contrastive learning strategy, with a view-specific fine-tuning network for handling incomplete data. This paper initially received four negative scores, and significant concerns remain regarding the paper's novelty and depth of analysis. After careful consideration of the reviews, the author rebuttal, and the ensuing discussion, the consensus is that the contribution does not meet the high bar for acceptance at ICLR.

**Reviewer Concerns:**

The core criticism from multiple reviewers is that the proposed framework is perceived primarily as a combination of existing ideas rather than a fundamental methodological advance. As noted by Reviewer etbz, the novelty appears combinatorial. Reviewer m1hA provides a pointed analysis, arguing that: (a) the dendritic fusion mechanism may degrade to a simple average in common scenarios (e.g., two views) and its empirical contribution in ablation studies is marginal; (b) key components (global contrast, parts of the local loss) have strong precedents in recent literature; and (c) consequently, the performance gains might be largely attributable to a specific sub-component (the redundancy loss) rather than the integrated framework. The rebuttal did not fully dispel this perception. For ICLR, where significant and clear conceptual advancements are prioritized, this presents a major weakness.

**Reviewer Scores:**

Both Reviewer etbz and m1hA clearly stated that rebuttal failed to address the core concerns and had no intention of raising the score

---

### Decision · Program_Chairs · 2026-01-26

Reject